# Private Federated Multiclass Post-hoc Calibration

## Abstract

Calibrating machine learning models so that predicted probabilities better reflect the true outcome frequencies is crucial for reliable decision-making across many applications. In Federated Learning (FL), the goal is to train a global model on data which is distributed across multiple clients and cannot be centralized due to privacy concerns. FL is applied in key areas such as healthcare and finance where calibration is strongly required, yet federated private calibration has been largely overlooked. This work introduces the integration of post-hoc model calibration techniques within FL. Specifically, we transfer traditional centralized calibration methods such as histogram binning and temperature scaling into federated environments and define new methods to operate them under strong client heterogeneity. We study (1) a federated setting and (2) a user-level Differential Privacy (DP) setting and demonstrate how both federation and DP impacts calibration accuracy. We propose strategies to mitigate degradation commonly observed under heterogeneity and our findings highlight that our federated temperature scaling works best for DP-FL whereas our weighted binning approach is best when DP is not required.

## 1 Introduction

Federated Learning (FL) is a decentralized approach to machine learning that allows models to be trained across distributed devices without requiring private data to be collected in a single location (McMahan et al., 2016; Kairouz et al., 2019). Rather than sending raw data to a server, clients perform local work and share model updates, such as gradients. This paradigm enables organizations to train models without having users send sensitive data and is well-suited to distributed environments where privacy is critical, such as mobile applications (Hard et al., 2018; Xu et al., 2022), healthcare systems (Xu et al., 2021), and financial services (Long et al., 2020). However, if applied in isolation, FL is not sufficient to protect the privacy of client data since it is vulnerable to attacks that leak sensitive information (Fowl et al., 2021). As such, it is commonly combined with Differential Privacy (DP) which adds carefully calibrated statistical noise into the training process (Dwork et al., 2006).

Most work in the federated setting has focused on aspects of model training, such as advancing federated optimization algorithms (Wang et al., 2019; Karimireddy et al., 2020; Wu et al., 2020), or improving the communication efficiency of training (Alistarh et al., 2017; Chen et al., 2021). However, in practical scenarios, model training is a small part of the overall ML pipeline. In many cases, federated deployments require additional solutions for the monitoring and improvement of existing models such as feature selection (Banerjee et al., 2021), computing federated evaluation metrics (Cormode & Markov, 2023) and post-processing for fairness (Chen et al., 2023).

In this work, we focus on the task of *post-hoc classifier calibration in the federated setting*. Calibration refers to the process of adjusting the output probabilities of a model to better reflect the true likelihood (confidence) of the predicted outcomes. Many applications of FL need to be able to trust model predictions as reliable estimates and thus require calibration: medical diagnosis, risk assessment scenarios (e.g., insurance) and finance (e.g., fraud detection). However, it is observed that modern neural networks (NNs) are typically not well-calibrated. That is, they are often extremely over or under-confident in their predictions (Guo et al., 2017).

We focus on calibration that is performed post-hoc once the model is trained. A post-hoc calibration model takes the output probabilities (or logits) of a trained classifier and learns a map to produce calibrated probabilities. One of the simplest approaches to (non-federated) post-hoc calibration is

Table 1: Our work compared to related private or federated calibration literature.

| Name | Federated | Differential Privacy | Multiclass | cwECE | Non-IID |
|------|-----------|---------------------|------------|-------|---------|
| Cormode & Markov (2023) | ✓ | ✓(Example-level) | ✗ | ✗ | ✗ |
| Bu et al. (2023) | ✗ | ✓(Example-level) | ✓ | ✓ | ✗ |
| Peng et al. (2024) | ✓ | ✗ | ✓ | ✗ | ✓ |
| Our Work | ✓ | ✓(User-level) | ✓ | ✓ | ✓ |

*histogram binning* (Zadrozny & Elkan, 2002). More sophisticated calibration methods involve scaling the final output logits of the neural network such as *temperature scaling* (Guo et al., 2017). Further extensions adapt scaling approaches to be better suited to multi-class classification (Kull et al., 2019; Zhao et al., 2021). Some central calibration methods integrate directly into model training (Mukhoti et al., 2020; Jung et al., 2023; Marx et al., 2024). However, we study post-hoc calibration which is more flexible, and allows calibrating pre-trained models to data in federated settings.

The challenge of federated calibration, with and without Differential Privacy (DP), remains largely unexplored. Bu et al. (2023) examine model calibration in the central setting and demonstrate that training models with DP exacerbates calibration error, highlighting the importance of calibration for private models. Peng et al. (2024) investigate scaling methods for a federated setting under strong data heterogeneity but crucially do not address how to incorporate DP. The most closely related work is due to Cormode & Markov (2023), who motivate DP-FL calibration. They describe a private histogram binning approach, but their method is restricted to binary classifiers and operates under *example-level DP*. In contrast, federated models are typically trained via the stricter notion of *user-level DP* and are often multi-class, which is the more challenging version to tackle. Their work does not address the impact of heterogeneity, which we study and design novel techniques for.

**Our contributions.** We propose novel methods to perform model calibration in federated heterogeneous environments, analyzing both private and non-private variants. For multiclass tasks, measures such as expected classwise error (ECE) can oversimplify the problem (Kull et al., 2019), so we seek to minimize the *classwise* calibration error (cwECE) for multi-class classification, with emphasis on ensuring user-level Differential Privacy (DP) in the private setting. Table 1 summarizes our work in comparison to related literature. We are the first to handle cwECE in the most challenging non-IID user-level DP-FL settings, in contrast to prior studies that study IID binary classifiers or omit DP. Our main findings are as follows:

**1.** We show naively applying existing methods to non-IID FL settings does not solve the problem, causing model accuracy to decrease after calibration. We propose two novel frameworks for federated post-hoc calibration which alleviate these issues, **FedBinning** (Alg 1) and **FedScaling** (Alg 2).

**2.** In the FL setting, motivated by the failures of naive methods, we modify federated binning via a weighting scheme that is applied by the server with no additional overhead. We conclude that our tailored enhancements outperform scaling approaches under strong heterogeneity. Extending our analysis to the DP-FL setting, we create user-level DP variations and demonstrate that our proposed scaling methods outperform binning due to the noise and clipping required for user-level DP binning.

**3.** Our findings are validated by experiments on seven benchmark datasets under two forms of data heterogeneity. We see that our approaches perform calibration effectively in heterogeneous federated settings. Our code is available at `https://anonymous.4open.science/r/priv_fed_calibration-5414/`

## 2 RELATED WORK

**Model Calibration.** Classifier calibration is a well-studied area for neural networks. Guo et al. (2017) highlight that common model architectures tend to be extremely over or under-confident in their predictions. Many approaches to post-hoc calibration exist, including classical approaches such as Platt scaling (Platt et al., 1999), isotonic regression (Chakravarti, 1989) and histogram binning (Zadrozny & Elkan, 2002) with extensions such as Bayesian averaging via BBQ (Naeini et al., 2015). Guo et al. (2017) show that temperature scaling, a simple approach that scales the output logits of the neural network, greatly reduces calibration error against a variety of post-processing calibrators . A recent line of work proposes an alternative to post-hoc calibrators via train-time augmentations to the training loss. These modifications usually include a regularizer that aims to minimize a proxy for

calibration error during model training. Examples include Mukhoti et al. (2020) who use the focal loss and Marx et al. (2024) who add a regularizer based on Maximum Mean Discrepancy (MMD). They also show that combining both a post-hoc calibrator (i.e., temperature scaling) with train-time augmentations can achieve best calibration error. We consider private train-time methods out of scope, as there are no DP-FL approaches and our focus is strictly on how to federate post-hoc calibrators.

**Privacy and Calibration.** DP is known to cause properties of NNs to worsen, such as accuracy (Sander et al., 2023) and fairness (Cummings et al., 2019; Bagdasaryan et al., 2019), where the noise addition and clipping from DP-SGD (Abadi et al., 2016) can worsen biases already present in the dataset. The only prior work that studies central privacy and its effects on calibration is that of Bu et al. (2023). They show models trained via DP-SGD have increased calibration error due to the noise addition and gradient clipping. They propose a centrally private temperature scaling algorithm trained via DP-SGD as an extra post-processing step after model training, and demonstrate this effectively lowers the calibration error.

**Federated Calibration.** Few works have studied model calibration in FL. Luo et al. (2021) consider the problem of developing prototypes to calibrate local loss functions to avoid heterogeneity and improve model convergence. Lee et al. (2024) apply logit chilling in the federated setting via local temperature scaling as a way to improve convergence speeds. Crucially, both works do not explicitly measure the calibration error of the resulting federated models. Recently, Peng et al. (2024) looks at post-processing calibrators in a heterogeneous federated setting. They propose FedCal, a scaling approach that trains a local multilayer perceptron (MLP). They evaluate the top-label ECE of the final model whereas we argue that classwise-ECE is better suited for federated multi-class problems and study the problem under privacy. Last, Chu et al. (2024) regularize the local loss function for calibration in federated training. Their approach relies on estimating the similarity between local and global models. In contrast to our work, they do not study post-hoc calibrators or private calibration.

**DP-FL Calibration.** Calibration under both DP and FL is mostly unexplored. The closest work is by Cormode & Markov (2023) who study model calibration and evaluation metrics in DP-FL. However, they restrict to binning methods for binary classifiers, with example-level DP and do not study client heterogeneity (i.e., from label-skew) and client subsampling. In contrast, we focus on multi-class classifiers calibrated in a heterogeneous federated setting with user-level DP.

## 3 PRELIMINARIES

**Differential Privacy & Federated Learning.** We assume a horizontal federated setting where $K$ distributed participants (e.g., mobile devices) each have a local dataset $D_1, \ldots, D_K$ with the full dataset denoted $D := \cup_k D_k$. We also assume the local datasets exhibit some form of heterogeneity, i.e., the datasets are not IID. We describe how we model this empirically in Section 5. We consider a multi-class classification setting where client $k$ has $n_k$ data samples with $d$ features and a target variable $y_i^k \in \mathcal{Y}$ where $|\mathcal{Y}| = c$ is the total number of classes. We are interested in training a global model $\theta(\boldsymbol{x}; \boldsymbol{w})$ with learned weights $\boldsymbol{w}$ to make predictions of the form $f(\boldsymbol{x}; \boldsymbol{w}) = \sigma(\theta(\boldsymbol{x}; \boldsymbol{w}))$ where $\sigma$ is the softmax function. Models are trained in the federated setting using algorithms such as FedAvg (McMahan et al., 2017a). At step $t$ of FedAvg training, participants are subsampled with probability $p$ to train their local model with the current global model weights $\boldsymbol{w}^t$ via local SGD for a number of epochs to produce client weights $\boldsymbol{w}_k^t$. This is sent back to the server which updates the global model via[1] $\boldsymbol{w}^{t+1} = \frac{1}{p \cdot K} \sum_k \boldsymbol{w}_k^t$ over all provided client weights at round $t$. This is often wrapped inside a lightweight cryptographic secure-aggregation protocol, SecAgg (Bonawitz et al., 2017), which allows the server to aggregate $\boldsymbol{w}^{t+1}$ without each client revealing their individual $\boldsymbol{w}_k^t$.

SecAgg alone is not enough to guarantee privacy on the output of federated computations (Fowl et al., 2021). Differential privacy (Dwork et al., 2006) is a formal definition which guarantees the output of an algorithm does not depend too heavily on any one individual. We aim to guarantee user-level $(\varepsilon, \delta)$-DP, where $\varepsilon$ is named the *privacy budget* and determines an upper bound on the privacy leakage of a differentially private algorithm. The parameter $\delta$ is set very small and corresponds to the probability of failing to meet the DP guarantee.

---

[1]This formula assumes the server learning rate is 1—see Appendix B.2.

**Definition 3.1** $((\varepsilon, \delta)$-DP$)$. *A randomized mechanism $\mathcal{M}$ is $(\varepsilon, \delta)$-differentially private if for any neighboring datasets $D, D'$ and any possible subset of outputs $S$ we have $\mathbb{P}(\mathcal{M}(D) \in S) \leq e^{\varepsilon}\mathbb{P}(\mathcal{M}(D') \in S) + \delta$*

We assume *user-level privacy*, which says two datasets $D, D'$ are adjacent if $D'$ can be formed as the addition/removal of a single user's data in $D$. In FL, to guarantee user-level DP, clients must clip contributions to ensure bounded sensitivity. For DP-FedAvg (McMahan et al., 2017b), the model updates sent from clients are clipped to have norm $C$ and therefore global sensitivity $C$. We show how to extend our methods to user-level DP in Section 4.3 with full details in Appendix B.1.

**Threat Model.** We assume an honest-but-curious model where participants do not trust others with their private data, including their model updates. In addition, we assume there is a central server that follows the algorithm exactly. We will use a secure-aggregation protocol (Bell et al., 2020) to allow clients and the server to aggregate local updates. When enforcing the use of DP, we assume the central server adds the necessary privacy noise to the aggregated output of secure-aggregation.

**Calibration.** We focus on calibration of neural networks for multi-class classification. The goal is to achieve perfect calibration of such a classifier.

**Definition 3.2** (Perfect Calibration). *Specifically, for labels $Y \in \mathcal{Y} = \{1, \ldots, c\}$ and a predictor $\hat{P}$ with confidence predictions $\hat{p}$ we define perfect calibration as $\mathbb{P}(\hat{Y} = Y | \hat{P} = \hat{p}) = \hat{p}, \forall \hat{p} \in [0, 1]$.*

As this requires knowledge of the underlying joint distribution it is impossible to achieve in practice. Instead we rely on empirical measurements of how well-calibrated a classifier is. For a single target class, the most common metric is the Expected Calibration Error (ECE) which partitions predicted confidences $\hat{p} \in [0, 1]$ into bins and averages the difference between accuracy and predicted confidence within each bin. Several works have studied alternative measures of calibration (Gupta & Ramdas, 2021; Błasiok & Nakkiran, 2023). We apply a stricter measure for the multiclass setting called classwise-ECE (cwECE) (Kull et al., 2019).

**Definition 3.3** (Classwise-ECE). *For a fixed partitioning of $[0, 1]$ into $B_1, \cdots, B_M$ bins, the cwECE is defined as $\frac{1}{c}\sum_{j=1}^{c}\sum_{m=1}^{M}\frac{|B_m|}{N}|\hat{p}_j(m) - y_j(m)|$ where $N$ is the total number of samples, $\hat{p}_j(m) := \frac{1}{|B_m|}\sum_{i \in B_m} \hat{p}_{i,j}$ is the average class $j$ confidence of examples in $B_m$ and $y_j(m)$ is the proportion of class $j$ examples in $B_m$.*

Our focus is on how to federate post-hoc calibrators. Here, the goal is to learn a map $g : \mathbb{R} \to [0, 1]$ as a form of post-processing over the predictions of a model (i.e., logits or probabilities) which outputs calibrated probabilities. The simplest approach is based on histogram binning for binary classification (Zadrozny & Elkan, 2002). The uncalibrated predicted probabilities $\hat{p}_i$ are partitioned into $M$ bins and each bin is remapped to a new confidence score. Further extensions use Bayesian Binning into Quantiles (BBQ) which averages multiple binning schemes to produce the final calibrator. See Appendix B.3 and B.4 for full details of these methods. More recently, Guo et al. (2017) proposed a class of calibrators known as scaling methods. These involve learning a linear transformation of the output logits of the neural networks (and not raw probabilities). Scaling approaches take the form $g(\boldsymbol{z}_i) = \sigma(A\boldsymbol{z}_i + \boldsymbol{b})$ where $\boldsymbol{z}_i$ is the output logits for example $\boldsymbol{x}_i$.

In our federated setting, we assume that each client partitions their local dataset $D_k$ into three sets; train, calibration and test. We measure the cwECE of the global model produced from federated training computed over these federated test sets as a measure of global cwECE. This is in contrast to Peng et al. (2024). who study a global form of top-label ECE. In practice, we only need clients to have a train and calibration set, which is easily achieved by splitting clients' local data into two.

## 4 FEDERATED CALIBRATION

We focus on two classes of methods: histogram binning and scaling. Section 4.1 highlights the issues with naively extending binning to FL and puts forward our weight aggregation strategy to alleviate this. Section 4.2 shows how to federate scaling methods, again overcoming difficulties due to heterogeneity. Section 4.3 proposes extensions to our framework for user-level DP.

---

**Algorithm 1** Federated Multiclass Binning

---

**Input:** Local datasets $D_1, \ldots, D_K$, sampling rate $p$, trained model $\theta$, $M$ bins, $T$ rounds, (optional) $(\varepsilon, \delta)$
**for** each round $t = 1, \ldots T$ **do**
    Server samples participants with probability $p$ to form the participation set $\mathcal{P}_t$
    Partition $[0, 1]$ into $M$ fixed-width bins, $B_m := \left[\frac{m-1}{M}, \frac{m}{M}\right]$
    **for** each client $k \in \mathcal{P}_t$ and each class $j \in [c]$ **do**
        Compute local histograms over positive and negative class $j$ examples with confidences $\{\hat{p}_{i,j}\}_{i \in D_k}$

$$P_j^k(m) := \sum_{i \in D_k} \mathbf{1}\{y_i = j \wedge \hat{p}_{i,j} \in B_m\}, \quad N_j^k(m) := \sum_{i \in D_k} \mathbf{1}\{y_i \neq j \wedge \hat{p}_{i,j} \in B_m\}$$

        Client $k$ sends each local class histogram $\{P_j^k, N_j^k\}$ to the server via SecAgg (Bell et al., 2020)
    Server aggregates classwise histograms $P_j = \sum_{k \in \mathcal{P}_t} P_j^k, N_j = \sum_{k \in \mathcal{P}_t} N_j^k$
    **Optional:** Server clips each $\{P_j, N_j\}$ and adds Gaussian noise to guarantee $(\varepsilon, \delta)$-DP, see Sec 4.3
    For each class $j \in [c]$, the server forms the one-vs-all binning calibrator of the form

$$g_j(\hat{p}) := \sum_m \mathbf{1}\{\hat{p} \in B_m\} \frac{P_j(m)}{P_j(m) + N_j(m)}$$

    Return $g(\hat{\boldsymbol{p}}_i) := (g_1(\hat{p}_{i,1}), \ldots, g_c(\hat{p}_{i,c})) / \sum_{j=1}^c g_j(\hat{p}_{i,j})$

---

**Algorithm 2** Federated Scaling

---

**Input:** $K$ participants with data $D_1, \ldots, D_K$, sampling rate $p$, $T$ calibration rounds, (optional) $(\varepsilon, \delta)$
**for** each global round $t = 1, \cdots, T$ **do**
    Server samples participants with probability $p$ to form the participation set $\mathcal{P}_t$
    **for** each client $k \in \mathcal{P}_t$ **do**
        Client trains a local scaling calibrator $g_k(\boldsymbol{z}) := \sigma(A\boldsymbol{z} + \boldsymbol{b})$ over their logits $\{\boldsymbol{z}_i\}_{i \in D_k}$
        Client sends final local scaling parameters $A_k, \boldsymbol{b}_k$ to the server via SecAgg (Bell et al., 2020)
    Server aggregates and averages scaling parameters, $A_t = \frac{1}{|\mathcal{P}_t|} \sum_{k \in \mathcal{P}_t} A_k, \boldsymbol{b}_t = \frac{1}{|\mathcal{P}_t|} \sum_{k \in \mathcal{P}_t} \boldsymbol{b}_k$
    **Optional:** Server clips $A_t, \boldsymbol{b}_t$ and adds DP noise calibrated to $(\varepsilon, \delta)$-DP over $T$ rounds, see Sec 4.3
Return final calibrator of the form $g(\boldsymbol{z}) = \sigma(A_T \cdot \boldsymbol{z} + \boldsymbol{b}_T)$

---

## 4.1 HISTOGRAM BINNING APPROACHES

Histogram binning is usually applied in a binary setting ($c = 2$ classes). The goal is to map the model's predicted probabilities to more accurate confidence estimates based on the actual outcomes observed. This process is carried out by partitioning the interval $[0, 1]$ into bins and replacing the scores of examples in a bin by the average accuracy over all examples within that bin. See Appendix B.3 for a formal treatment of (centralized) histogram binning. Cormode & Markov (2023) propose a translation of binning to the federated setting. Their idea is to build histograms of positive and negative examples over the bins. This yields two histograms built from client responses that are combined via secure-aggregation (Bell et al., 2020). Once the server receives these histograms it can build the calibrator by computing the accuracies within each bin.

**Multiclass binning technique.** We turn the problem into $c$ one-vs-all calibrators. That is, we learn a binning calibrator $g_j$ for each class $j$. To do so, each client computes and sends $2c$ histograms—the positive and negative histograms for each class. The server receives the sum of all these histograms, which define $c$ binary binning calibrators. The final prediction is the normalized probability distribution from each one-vs-all calibrator. We present the framework for federated binning in Algorithm 1.

Directly implementing this approach has some highly undesirable properties in the multi-class federated setting. We refer to this naive baseline as FedBin. To demonstrate the drawbacks, we applied FedBin to calibrate a simple CNN trained over CIFAR10 with significant label-skew (see Section 5 and Appendix A.2 for full details) with 100 clients and 10 clients participating per-round. Figure 1a shows that the baseline binning approach cannot simultaneously achieve low cwECE and good accuracy for high skew (small $\beta$). Figure 1b shows the cause of this failure mode. It plots the change in average training accuracy on clients' local datasets after they perform local calibration next to the training accuracy change on the global dataset. Letting clients train local calibrators when they have significant skew causes overfitting: a big *increase* in (local) training accuracy but a large

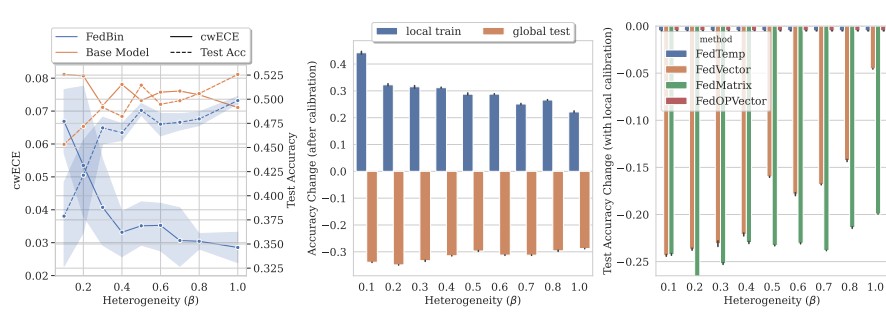

(a) Binning: cwECE & test acc (b) Binning: Local predictions (c) Scaling: Local predictions

Figure 1: Naive federated calibration on CIFAR10 (Simple CNN), varying heterogeneity $\beta$.

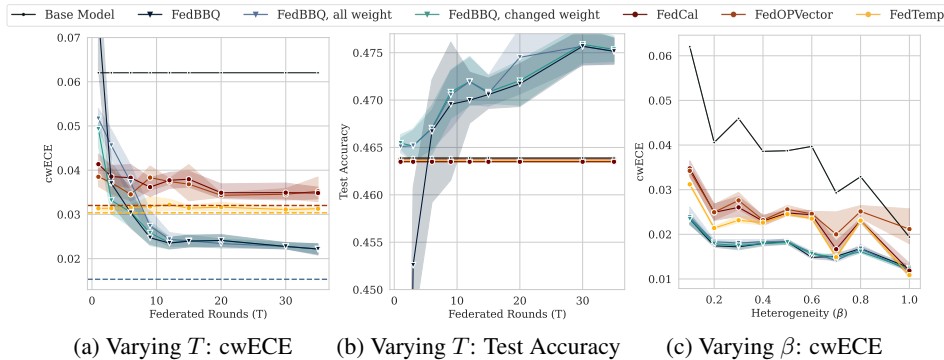

(a) Varying $T$: cwECE     (b) Varying $T$: Test Accuracy     (c) Varying $\beta$: cwECE

Figure 2: FL Calibration on CIFAR10 (Simple CNN), $\beta = 0.1$ unless otherwise stated.

*reduction* in (global) test accuracy. This overfitting causes significant problems when the federated binner is trained over very few rounds, creating a global calibrator that significantly decreases the overall test accuracy, which is an unacceptable tradeoff.

**Addressing Heterogeneity.** Our solution to this overfitting problem is for the server to use a weighting scheme when combining the client histograms. The high-level idea is to have the server output a calibrator whose prediction is a weighted average between the base model's confidence and that of the federated binning calibrator for each class $j$. We define the weighted calibrator as $\tilde{g}_j(\hat{p}) = \alpha_j \cdot g_j(\hat{p}) + (1 - \alpha_j) \cdot \hat{p}$, where $\alpha_j = \text{clip}(\frac{\tilde{N}_j}{N_j}, 1)$, $\tilde{N}_j$ is the total number of class $j$ examples aggregated so far and $N_j$ is the total number of class $j$ examples in the dataset. This weighting factor, $\alpha_j$, balances reliance on the federated calibrator, trusting the calibrator only when there is a sufficient number of class $j$ samples that have been observed, ensuring stability under label skew or a small number of calibration rounds.

**Instantiating Algorithm 1: FedBBQ.** The last part of our solution is for the server to combine information at multiple bin granularities. In the central setting, the best binning-based calibrator, BBQ, fuses multiple different histograms, by applying Bayesian averaging (Naeini et al., 2015). We instantiate this averaging in the federated model by materializing a collection of histograms with different bin counts. Specifically, we build a histogram with $2^M$ total bins (via Algorithm 1) and form $M$ different histograms by merging bins. This forms $M$ calibrators which are averaged using the Bayesian weighting of BBQ to produce a final calibrator. We apply our weighting scheme for addressing heterogeneity to the final output. We refer to the final combined protocol as FedBBQ, and give full details in Appendix B.4. Subsequently, we only show results for FedBBQ, as it outperformed the single unweighted histogram binning method FedBin. We fix $M = 7$ (128 total bins) as this has consistent results across all settings, based on our ablation study (summarized in Appendix A.3.3).

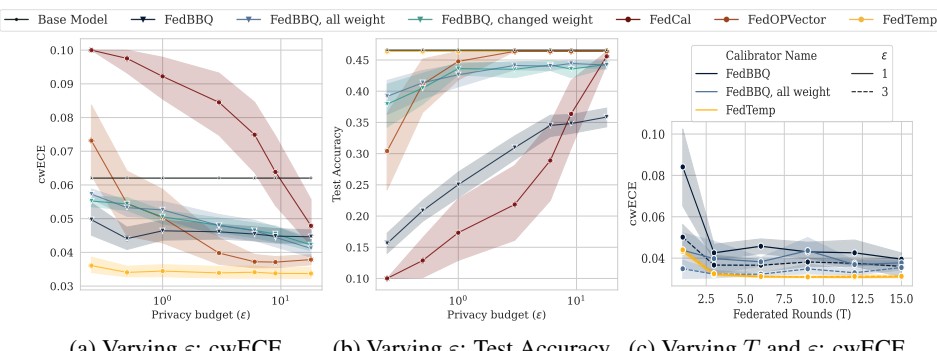

(a) Varying $\varepsilon$: cwECE  (b) Varying $\varepsilon$: Test Accuracy  (c) Varying $T$ and $\varepsilon$: cwECE

Figure 3: DP-FL Calibration on CIFAR10 (Simple CNN) $\beta = 0.1$ varying $\varepsilon$ with $\delta = 10^{-5}$

## 4.2 SCALING APPROACHES

We next design federated versions of the scaling approaches introduced by Guo et al. (2017). This learns a linear transformation of the output logits $g(\boldsymbol{z}_i) = \sigma(A\boldsymbol{z}_i + \boldsymbol{b})$. Since scaling calibrators are a transformation of logits, they can be viewed as a neural network layer and trained via a FedAvg framework. That is, we let clients train their local scaler and have them share the local parameters $A_k, \boldsymbol{b}_k$ which the server aggregates over multiple rounds. The global calibrator can be updated using the average of these parameters. This framework is outlined in Algorithm 2.

**Instantiating Algorithm 2.** We define the following scaling approaches: FedMatrix with $A, \boldsymbol{b}$ unconstrained; FedVector with $A$ restricted to diagonal entries; and FedTemp where $A = \mathbf{1}/a, b = 0$.

However, similar problems arise when naively applying scaling methods as with naive binning. Figure 1c shows the change in (global) test accuracy after local scaling to highlight issues on non-IID data. As heterogeneity increases ($\beta$ decreasing), naive vector (FedVector) and matrix (FedMatrix) scaling suffer an increasing penalty to the global accuracy. This local overfitting results in final calibrated models with worse test accuracy than the base model. As vector and matrix scaling learn a bias term $\boldsymbol{b}$, this shifts class-predictions and changes accuracy. There is no accuracy loss for FedTemp (temperature scaling) as it only rescales logits, preserving the relative ordering of model predictions.

**Addressing Heterogeneity.** To prevent accuracy loss, we can enforce that the scaling for calibration does not change the predictions by adopting *order-preserving training* (Rahimi et al., 2020). Combined with vector scaling, this ensures that the top-k accuracy is unchanged and so no accuracy degradation occurs. We denote this approach as FedOPVector. In Figure 1c, we see that FedOPVector does not suffer any accuracy loss. In our later experiments, we compare to the FedCal approach from Peng et al. (2024) which uses an MLP to transform the logits, also in combination with order-preserving training [2]. We present more details on order-preserving training in Appendix B.6.

## 4.3 FEDERATED CALIBRATION WITH USER-LEVEL DP

Our methods provide DP guarantees by appropriate noise addition and clipping during a federated round of calibrator training. We guarantee user-level DP by clipping each user's contributions based on a parameter $C$, ensuring bounded sensitivity. Appendix B.1 gives our formal $(\varepsilon, \delta)$-DP guarantees.

**Binning under user-level DP.** As federated binning approaches require user-level DP histograms, we apply the methods of Liu et al. (2023). One issue we observe is that histograms are particularly sensitive to the chosen clipping norm $C$. Such sensitivity to $C$ is exacerbated in histogram binning as the positive and negative example histograms are often over different scales, e.g., there are far more negative examples than positive in a one-vs-all setting. We overcome this by using two separate clipping norms $C_+, C_-$ which determine the clipping norm for the individual histograms. A different problem occurs when utilizing FedBBQ over multiple binning schemes, as the privacy cost scales with the number of binners. Our solution avoids this, as we instantiate a single large histogram of size $2^M$ and form multiple binning schemes by merging bins. Under these settings, the privacy cost

---

[2]They propose an additional scheme in a train-time calibration setting which we omit in our post-hoc setting.

Table 2: Comparison of cwECE across methods in a non-DP FL setting, $\beta = 0.1$ for non-LEAF datasets and $T = 12$. Mean cwECE is reported as a % alongside standard deviation over 5 runs. Results marked $^*$ suffer $> 1\%$ drop in test accuracy after calibration.

| Dataset | Base | FedBBQ | FedBBQ All weight | FedCal | FedOPVector | FedTemp |
|---|---|---|---|---|---|---|
| CIFAR10 (18) | 8.11 | **2.375 (0.276)** | 2.499 (0.36) | 4.391 (0.849) | 3.779 (0.741) | 2.428 (0.112) |
| CIFAR100 (18) | 0.94 | 0.317$^*$ (0.017) | **0.29 (0.02)** | 0.526 (0.025) | 0.472 (0.018) | 0.341 (0.01) |
| SVHN (18) | 2.05 | **1.46 (0.145)** | 1.483 (0.247) | 2.468 (0.165) | 2.479 (0.51) | 2.172 (0.182) |
| Tinyimnet (18) | 0.21 | 0.211$^*$ (0.005) | 0.194 (0.002) | 0.187 (0.001) | **0.187 (0.0)** | 0.198 (0.001) |
| FEMNIST (2) | 0.18 | 0.161 (0.005) | **0.153 (0.004)** | 0.237 (0.048) | 0.205 (0.014) | 0.182 (0.001) |
| MNIST (2) | 0.77 | **0.557 (0.037)** | 0.586 (0.046) | 1.232 (0.238) | 0.865 (0.202) | 0.763 (0.018) |
| Shakespeare (3) | **0.14** | 0.212 (0.001) | 0.2 (0.002) | 0.331 (0.06) | 0.38 (0.04) | 0.165 (0.052) |

Table 3: DP-FL comparison, $\beta = 0.1$ for non-LEAF datasets, $T$=12 and $(\varepsilon, \delta) = (1, 10^{-5})$. Methods in bold achieve lowest cwECE for a particular dataset, ignoring methods that suffer accuracy drops.

| Dataset | Base | FedBBQ | FedBBQ All weight | FedCal | FedOPVector | FedTemp |
|---|---|---|---|---|---|---|
| CIFAR10 (18) | 8.11 | 4.582$^*$ (1.03) | **3.086 (0.15)** | 9.999$^*$ (0.002) | 4.118 (0.812) | 4.423 (0.122) |
| CIFAR100 (18) | 0.94 | 0.129$^*$ (0.005) | 0.132$^*$ (0.008) | 1.0$^*$ (0.0) | 0.592 (0.033) | **0.346 (0.006** |
| SVHN (18) | 2.05 | 6.89$^*$ (0.64) | 2.047 (0.002) | 9.955$^*$ (0.034) | 2.396 (0.528) | **2.007 (0.025)** |
| Tinyimnet (18) | 0.21 | 0.232$^*$ (0.017) | 0.212 (0.0) | 0.5$^*$ (0.0) | **0.192 (0.009)** | 0.198 (0.003) |
| FEMNIST (2) | 0.18 | 1.888$^*$ (0.029) | 0.184 (0.002) | 0.232 (0.025) | 0.201 (0.009) | **0.18 (0.001)** |
| MNIST (2) | 0.77 | 9.381$^*$ (0.86) | 0.769 (0.0) | 9.998$^*$ (0.003) | 2.607 (2.153) | **0.746 (0.008)** |
| Shakespeare (3) | 0.14 | 1.281$^*$ (0.051) | 0.15 (0.003) | 0.387 (0.121) | 0.339 (0.057) | **0.138 (0.0)** |

scales proportional to $2Tc$ where $c$ is the number of classes and $T$ is the total number of federated calibration rounds (see Appendix B.1).

**Scaling under user-level DP.** As the scaling methods are trained under FedAvg, we can apply DP-FedAvg with minimal changes (McMahan et al., 2017b). Here, the local scaling parameters $A_k, \boldsymbol{b}_k$, are clipped to norm $C$ by clients and shared with the server under secure-aggregation (Bell et al., 2020). The server can then average these aggregated parameters and add calibrated Gaussian noise to guarantee user-level DP. Here, the privacy cost scales proportional to $T$, the total number of federated calibration rounds.

## 5 EXPERIMENTS

We perform federated calibration on a base model trained via FedAvg without DP on 7 datasets: MNIST with a 2-layer CNN and ResNet18; CIFAR10/100 with a 2-layer CNN and ResNet18; SVHN, Tinyimagenet and FEMNIST with ResNet18; and Shakespeare with an LSTM for next-character prediction. See Appendix A.1 and A.3 for dataset and model training details. For FEMNIST and Shakespeare we utilize the LEAF benchmark (Caldas et al., 2018) which federates in a natural way to induce heterogeneity. For other datasets we assume $K = 100$ clients and simulate heterogeneity via the label-skew approach of Yurochkin et al. (2019). This involves sampling client class distributions from a Dirichlet($\beta$) where smaller $\beta$ creates more skew (see Appendix A.2). Figures on other datasets can be found in Appendix A.4. Our FL models achieve test accuracy that matches or improves over those used in FedCal (Peng et al., 2024), see Table 7. Our code is also open-sourced.

For instantiating our binning framework (Alg 1) we consider FedBBQ and two variations with our weighting-scheme: *all weight*, applies the weighting scheme to all predictions and *changed weight* which only applies weights to examples that have their class prediction changed by the calibrator. For scaling (Alg 2), we consider FedTemp and FedOPVector. We also compare to a post-hoc variant of FedCal (Peng et al., 2024). We focus on cwECE in our experiments, see Appendix A.5 for a detailed discussion on why this is preferable to ECE. We are interested in answering the following research questions:

- **RQ1 (Non-IID FL)**: Which method is most effective in a non-IID FL setting?

- **RQ2 (Mitigating Heterogeneity)**: Do our proposed mitigations for heterogeneity prevent accuracy loss in both FL and DP-FL settings?

- **RQ3 (DP-FL)**: Which method achieves the best balance of accuracy and calibration error in a non-IID user-level DP-FL setting?

- **RQ4 (Robustness):** How robust are these findings across different datasets?

**RQ1 (Non-IID FL):** In Figure 2a, we vary the number of federated rounds ($T$) used to train the calibrator on CIFAR10 with no DP. We observe FedTemp has consistently good cwECE but that as $T$ increases the FedBBQ approach outperforms it. This is because binning methods aggregate histograms, and, if given enough rounds, can closely match the performance of a central calibrator (displayed in dashed lines). Note that while FedBBQ only outperforms scaling when $T$ is around 30, this is far smaller than the total number of federated rounds required to train the base FL model itself. Hence, these calibration approaches are lightweight and can be performed as an extra federated epoch at the end of training with modest additional overhead to the overall federated training pipeline. We find scaling methods with a larger number of parameters (i.e., FedCal) often perform the worst. **Summary:** For the FL setting, FedTemp is the most competitive scaling method when $T$ is small but is outperformed by FedBBQ when trained over multiple rounds ($T > 30$). We recommend using FedBBQ with a moderate number of rounds ($T = 30$) in the non-DP FL setting.

**RQ2 (Mitigating Heterogeneity):** In Figure 2c, we vary $\beta$ on CIFAR10. We fix the number of FL rounds to $T = 12$. We observe binning approaches work well under all ranges of heterogeneity as the algorithm aggregates histograms which mostly mitigates skew. We observe FedTemp achieves best cwECE out of the scaling approaches but is outperformed by binning methods. In Figure 2b, we fix $\beta = 0.1$ and vary $T$. We observe all scaling methods obtain the same accuracy as the base model (all scaling methods in the figure are overlaid). We clearly find our weighting scheme for FedBBQ and the order-preserving scaling methods prevent any loss in test accuracy. In contrast, if the number of rounds is too small, unweighted FedBBQ suffers a large drop in accuracy as discussed in Section 4.1. **Summary:** Our weighting schemes prevent accuracy degradation for FedBBQ and the order-preserving training of FedOPVector prevents the accuracy degradation observed in FedVector in Section 4. FedBBQ with weighting has better ECE than FedOPVector and so this is further evidence to recommend the use of weighted FedBBQ for non-DP FL settings.

**RQ3 (DP-FL):** In Figure 3 we study the DP-FL setting, varying the privacy budget ($\varepsilon$) and compare binning to scaling. We set $C = 0.5$ for scaling methods and $C = [10, 50]$ for binning approaches. Observe FedCal, which utilizes an MLP, is the most sensitive to noise since it has the most model parameters. For any reasonable privacy ($\varepsilon < 10$) it achieves larger cwECE than the base model. We find FedTemp is the most resilient to high noise since it shares only a single parameter under DP-FedAvg. Binning methods struggle under DP-noise due to the user-level clipping on histograms. We explore this further in Appendix A.4.2. In Figure 3c, we vary both the number of rounds ($T$) and the privacy budget ($\varepsilon$) for CIFAR100 (CNN) and plot the cwECE. We study $\varepsilon = 1, 3$ for FedTemp, FedBBQ and our weighted FedBBQ. We find FedTemp is insensitive to DP noise, with a decrease in cwECE that plateaus quickly as $T$ increases. In contrast, the binning approaches often struggle in a one-shot ($T = 1$) setting but closely match FedTemp as $T$ increases, even under DP noise. **Summary:** In the DP-FL setting, we recommend using FedTemp as it achieves the best cwECE with no accuracy degradation unlike FedBBQ methods which may degrade accuracy under high privacy.

**RQ4 (Robustness):** In Tables 2 and 3 we explore the cwECE across each benchmark dataset in FL and DP-FL settings. For non-DP (Table 2), we observe binning approaches consistently outperform scaling methods, achieving lowest cwECE overall. On two datasets, FedBBQ degrades model accuracy, something our weighting scheme helps prevent. For DP-FL (Table 3), we find FedTemp consistently achieves best cwECE with smaller variance. It is clear that under DP noise methods with fewer parameters perform best (i.e., FedTemp) compared to FedCal or FedBBQ which have a larger number of parameters. We note FedTemp with DP sometimes achieves better cwECE than without DP. We find the clipping involved in DP actually helps mitigate local skew and show this empirically in Appendix A.3.5. We also flag results that suffer a drop of $> 1\%$ test accuracy. This reveals the only method to suffer a severe drop is the naive unweighted FedBBQ approach (with and without DP), highlighting the necessity of our weighting schemes which prevent accuracy degradation. **Summary:** Extending our results across multiple datasets, we find FedBBQ remains best in a FL setting under strong heterogeneity. Conversely, for DP-FL we find FedTemp achieves the best balance of accuracy and cwECE. We find large-parameter scalers like FedCal unsuitable in all settings.

Table 4: Federated post-hoc vs. train-time calibration on CIFAR10, varying the heterogeneity parameter $\beta \in \{0.1, 0.5, 1\}$. Each entry displays test accuracy and cwECE.

| Method / $\beta$ | 0.1 | 0.5 | 1.0 (IID) |
|---|---|---|---|
| Base Model (FedAvg) | 0.479 / 5.962 | 0.565 / 3.473 | 0.579 / 2.339 |
| *Train-time only.* | | | |
|     MMCE (Kumar et al., 2018) | 0.466 / 7.215 | 0.552 / 3.606 | 0.587 / 2.417 |
|     Focal (Mukhoti et al., 2020) | 0.411 / 5.175 | 0.569 / 1.944 | 0.560 / 1.558 |
|     MMD (Marx et al., 2024) | 0.468 / 6.155 | 0.559 / 3.476 | 0.592 / 2.032 |
|     NUCFL (Chu et al., 2024) | 0.493 / 5.830 | 0.563 / 3.029 | 0.597 / 1.990 |
| *Post-hoc only.* | | | |
|     FedTemp | 0.479 / 3.590 | 0.565 / 2.509 | 0.579 / 1.577 |
|     FedBBQ (All weight) | 0.503 / 2.382 | 0.576 / 1.893 | 0.578 / 1.332 |
| *Train-time + Post-hoc.* | | | |
|     NUCFL + FedTemp | 0.493 / 3.161 | 0.563 / 1.708 | 0.597 / 1.173 |
|     NUCFL + FedBBQ (All weight) | 0.524 / 2.286 | 0.562 / 1.564 | 0.597 / 1.262 |

Table 5: Client overheads for federated calibration on CIFAR10 (ResNet18), $c$ is the number of classes, $N$ is the number of hidden neurons in the FedCal MLP and $M$ is the FedBBQ bin parameter.

| Calibrator | Communication | Send Size (per-round) | Receive Size (per-round) | Model Size (received once) | Client Time (mean) |
|---|---|---|---|---|---|
| FedTemp | $O(1)$ | 0.01kb | 0.01kb | 43678.29kb | 0.124s |
| FedOPVector | $O(c)$ | 0.16kb | 0.16kb | 43678.29kb | 0.026s |
| FedCal | $O(N^2 + c^2)$ | 43.08kb | 43.08kb | 43678.29kb | 0.015s |
| FedBBQ, all weight | $O(2^M c)$ | 20.0kb | 0.0kb | 43678.29kb | 0.049s |

**Post-hoc vs. Train-time calibration.** In Table 4, we report train-time calibration results on CIFAR10 as we vary the label-skew parameter $\beta$. Each train-time method was originally proposed in the central setting, and we adapt its loss function for use within FedAvg. Concretely, each method optimizes a local loss of the form $\mathcal{L}(x_i, y_i) = \ell(x_i, y_i) + \alpha \, \ell_{\text{cal}}(x_i, y_i)$, where $\ell$ is the standard cross-entropy loss and $\ell_{\text{cal}}$ is a regularization term that serves as a proxy for calibration error. For all methods, we use the values of $\alpha$ recommended in the original works. The one exception is NUCFL (Chu et al., 2024), a federated train-time method that employs an adaptive weighting scheme in which $\alpha$ is set by each client, according to the similarity between global and local model updates; we use their cosine-similarity variant. Overall, train-time calibration exhibits very inconsistent behavior: while some methods can modestly improve accuracy over standard FedAvg, they often increase cwECE. In contrast, our post-hoc methods, which calibrate the predictions of the base FedAvg model, consistently outperform all train-time approaches in both accuracy and cwECE. The best performance is achieved when combining train-time and post-hoc calibration, aligning with observations from prior work (Marx et al., 2024; Chu et al., 2024). These results highlight the importance of post-hoc calibration to attain strong overall federated performance.

**Client Overheads.** Table 5 reports the per-round communication cost and local computation time for clients. These results are obtained in a simulated setting, so they exclude any communication overhead associated with secure-aggregation. Instead, the benchmarks reflect only the raw cost of transmitting calibration model parameters between the server and clients. Across all methods, computation is minimal, with local processing taking less than one second. Communication is very lightweight: receiving the global model from the server incurs roughly 500× more cost than participating in a single round of calibration in which the calibrator parameters are sent to the server. Our approaches remain lightweight compared to FedCal, whose calibrator model involves substantially more parameters. We provide full details for the communication complexity of each method in Appendix A.6.

**Conclusion.** Our results show we can calibrate effectively in the federated model, even in the face of high heterogeneity, and skew, without affecting model accuracy. Overall, for FL, we find binning with weight adjustments achieves best cwECE if trained over multiple rounds. However, in DP-FL, temperature scaling is preferred as it is resilient to both non-IID skew and DP noise, something binning and higher-order scalers struggle with. Thus, we can make clear recommendations for how best to calibrate models in each setting.

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

Table 6: Datasets used in our experiments.

| Dataset | Total Samples ($|D|$) | Number of clients ($K$) | Heterogeneity Type | Model Arch |
|---|---|---|---|---|
| MNIST | 70,000 | 100 | Label-skew | 2-layer CNN |
| CIFAR10 | 70,000 | 100 | Label-skew | ResNet18 |
| CIFAR100 | 70,000 | 100 | Label-skew | ResNet18 |
| SVHN | 70,000 | 100 | Label-skew | ResNet18 |
| Tinyimagenet | 100,000 | 100 | Label-skew | ResNet18 |
| FEMNIST | 805,263 | 1778 | LEAF | ResNet18 |
| Shakespeare | 4,226,15 | 607 | LEAF | LSTM |

# A EXPERIMENT DETAILS

All experiments (federated model training and federated calibration) were performed on a machine with a Dual Intel Xeon E5-2660 v3 @ 2.6 GHz and 64GB RAM. No GPUs were used in the training or calibration of the federated models. All experiments simulate client training sequentially (i.e., there is no paralellism) and individual federated calibration runs finish within 1 hour with full experiments (with 5 repeats) taking at most 24 hours.

## A.1 DATASETS AND MODELS

We use a variety of datasets in our experiments, as described in Table 6. In all experiments we merge the train and test sets to create a single global dataset. This dataset is then federated to clients via the label-skew approach described in Appendix A.2. Each client's local dataset is split into a train, test and calibration set. We take 80% of the dataset as training and 10% each for testing and calibration. All base models are also trained in a federated setting. In more detail:

- **MNIST** (Deng, 2012) is an image classification dataset that classifies handwritten numerical digits between 0-9. It has 10 classes. We train both a simple CNN and a ResNet18 model in our experiments. We partition the full dataset of 70,000 samples (train + test) across 100 clients. In the IID setting this results in each client having 560 train samples, 70 test and 70 calibration.

- **CIFAR10/CIFAR100** (Krizhevsky et al., 2009) is an image classification dataset with 10/100 classes. We train both a simple 2-layer CNN and a ResNet18 model in our experiments. We partition the full dataset of 70,000 samples (train + test) across 100 clients. In the IID setting this results in each client having 560 train samples, 70 test and 70 calibration.

- **SVHN** (Netzer et al., 2011) is an image classification dataset with 10 classes formed from house number digits. We partition the full dataset of $70,000$ samples across $100$ clients using the label-skew approach. In the IID setting, each client has 560 train samples, 70 test and 70 calibration.

- **Tinyimagenet (a.k.a. Tinyimnet)** (Le & Yang, 2015) is a subset of imagenet with 200 classes, 500 of each. We partition the full dataset using the label-skew approach but restrict $\beta \geq 0.3$ as setting $\beta$ very small can cause sampling issues in the label-skew partitioning procedure due to the large number of classes.

- **FEMNIST** (Caldas et al., 2018) is an image classification dataset with 62 classes. We use LEAF which provides a partition of the global dataset to clients where each client's local dataset has characters with similar handwriting to induce local heterogeneity. We use the "sample" partitioning which assigns user's local samples into train and test groups. We then take 50% of the local test set to form a client's calibration set. We filter away any users with fewer than 100 samples, resulting in 1,778 clients with an average of 182 samples for training, 23 for test and 23 for calibration.

- **Shakespeare** (Caldas et al., 2018) is a next-character prediction task trained on the works of Shakespeare. We use LEAF which provides a partition of the global dataset to induce heterogeneity. In this case, each speaking role, in each of Shakespeare's works, is a client. We use the "sample" partitioning which assigns user's local samples into train and test groups. We then take 50% of the local test set to form a client's calibration set. We filter

Table 7: Base model test accuracies for Simple CNN trained under FedAvg with $\beta \in [0.1, \ldots, 0.8]$

| Dataset / $\beta =$ | 0.1 | 0.2 | 0.3 | 0.4 | 0.5 | 0.6 | 0.7 | 0.8 | 1 (IID) |
|---|---|---|---|---|---|---|---|---|---|
| MNIST | 93.6% | 93.7% | 94.6% | 95.3% | 95% | 95.4% | 96.0% | 96.5% | 96.9% |
| CIFAR10 | 46.3% | 53.6% | 51.8% | 53.8% | 54.2% | 55.0% | 56.3% | 55.1% | 63.0% |
| CIFAR100 | 23.1% | 24.1% | 24.4% | 24% | 24.5% | 24.9% | 26.8% | 24.7% | 25.5% |

Table 8: Average number of unique classes in a clients' local dataset when varying label-skew heterogeneity ($\beta$).

| Dataset / $\beta =$ | 0.1 | 0.2 | 0.3 | 0.4 | 0.5 | 0.6 | 0.7 | 0.8 | 1 (IID) |
|---|---|---|---|---|---|---|---|---|---|
| MNIST | 4.05 | 5.65 | 6.59 | 7.11 | 7.49 | 7.9 | 8.05 | 8.32 | 10 |
| CIFAR10 | 3.94 | 5.38 | 6.37 | 6.94 | 7.53 | 7.73 | 8.1 | 8.2 | 10 |
| CIFAR100 | 23.91 | 28.76 | 30.97 | 33.48 | 34.63 | 35.97 | 36.94 | 37.5 | 45.22 |

away any users with fewer than 100 samples (i.e., roles with $< 100$ characters) resulting in 607 clients with an average of 5,590 samples for training, 293 for test and 293 for calibration.

We use the following model architectures:

- **Simple CNN:** We train a small convolutional neural network with 2 convolutional layers and 2 fully-connected layers. Our architecture follows that of the baseline used by Caldas et al. (2018)[3].

- **ResNet18:** We use the ResNet18 model on CIFAR100, SVHN, FEMNIST and Tinyimagenet in our experiments. This is consistent with its use by Peng et al. (2024) for federated calibration experiments.

- **Stacked LSTM**: For Shakespeare, we follow the model architecture used by Caldas et al. (2018) which is a stacked 3-layer LSTM. See the open-source implementation of LEAF for more details[4].

## A.2 MODELING HETEROGENEITY

In our experiments, we use two methods to partition our benchmark datasets into federated splits that exhibit heterogeneity.

**Label-skew**: In experiments where we vary heterogeneity we follow the approach outlined by Yurochkin et al. (2019). The label skew is determined by $\boldsymbol{\beta} = \beta \cdot \mathbf{1}^K$, where $K$ is the total number of clients. For each class value $j \in [c]$, we sample the client distribution via $\boldsymbol{p}_j \sim \text{Dirichlet}(\boldsymbol{\beta})$ where a smaller $\beta$ value creates more skew. For a particular client $k \in [K]$ we assign rows with class value $j$ proportional to $p_{j,k}$. This produces a partitioning of the dataset into clients that are skewed via the class values of the dataset. The parameter $\beta$ controls the strength of label-skew, where a larger $\beta$ decreases the skew and reduces heterogeneity. The setting of $\beta = 1$ corresponds to performing IID sampling for the client partitioning. In Table 8, we report the average number of unique classes present in each client's local dataset for different values of $\beta$. As expected, smaller values of $\beta$ induce substantially greater label skew, with the average client lacking most classes which makes local calibration difficult.

**LEAF:** For two of our datasets, FEMNIST and Shakespeare, we use a pre-partitioned split from the LEAF benchmark (Caldas et al., 2018). This federates data in a natural way to create heterogeneity in local datasets. As an example, FEMNIST is partitioned so that each user has digits written by the same hand.

---

[3]`https://github.com/TalwalkarLab/leaf/blob/master/models/femnist/cnn.py`
[4]`https://github.com/TalwalkarLab/leaf/blob/master/models/shakespeare/stacked_lstm.py`

Table 9: Client statistics for a single run of FedTemp on CIFAR10 whilst varying label-skew heterogeneity parameter ($\beta$). We present the variance across clients' local ECE using their local temperature parameter $T_k$, the variance across all client temperatures $T_k$ and the average absolute deviation between global optimal temperature $T$ and local $T_k$ i.e., $|T - T_k|$.

|  | var(local cwECE) | var($T_k$) | Mean($\|T - T_k\|$) |
|---|---|---|---|
| $\beta = 0.1$ | 0.069 | 1.39 | 1.07 |
| $\beta = 0.3$ | 0.039 | 0.468 | 0.524 |
| $\beta = 0.5$ | 0.023 | 0.264 | 0.432 |
| $\beta = 0.8$ | 0.022 | 0.249 | 0.419 |
| $\beta = 1$ (IID) | 0.0032 | 0.101 | 0.27 |

### A.3 Hyperparameters

#### A.3.1 Model Training

For all of our calibration experiments we calibrate a base model that has been trained on clients' local training datasets via FedAvg (McMahan et al., 2016). In datasets with label-skew, we train a model for each value of $\beta$ as this corresponds to a new partitioning of data to clients. In Table 7, we report the test accuracies of a federated simple CNN model on MNIST, CIFAR10 and CIFAR100 whilst varying heterogeneity ($\beta$). In general, models trained on federated splits with lower values of $\beta$ (corresponding to more local skew) have worse accuracy. We have comparable test accuracy to the models used by Peng et al. (2024) and so are suitable for federated calibration. For example, at $\beta = 0.1$ Peng et al. achieve 81% on MNIST (vs. ours at 94%), 48% on CIFAR10 (vs. ours at 46%) and 21% on CIFAR100 (vs. ours at 23%). We list the specific training hyperparameters used for each dataset and model architecture. Specifically, these are the local client learning rate $\eta_C$, the server learning rate $\eta_S$, the local batch size $B$ and number of global epochs $E$. In more detail:

**CIFAR10/100**: We use 100 clients and sample 10% per-round. We use a local client learning rate of $\eta_C = 0.01$ and a server rate of $\eta_S = 1$, and local batch size $B = 64$. For the simple CNN we train for $E = 300$ global epochs. For ResNet18 we train with $E = 100$ keeping the parameters the same as the above. We achieve an overall test accuracy of 63% in the IID case. For CIFAR100, the parameters are the same as CIFAR10 except the number of global epochs. For the simple CNN we set $E = 500$ and for ResNet18, $E = 100$. We achieve an overall test accuracy of 25% in the IID case.

**MNIST**: We train for $E = 10$ epochs with a local learning rate of $\eta_C = 0.001$ and server rate $\eta_S = 1$ and $B = 64$ for the simple CNN model. We achieve an overall test accuracy of 97% in the IID case.

**SVHN**: We use the same parameters as MNIST and train for $E = 10$ epochs with a ResNet18 model

**Tinyimagenet:** We use the same parameters as MNIST except train for $E = 30$ epochs.

**Shakespeare**: We take 60 clients per round. We train with $\eta_C = 0.01, \eta_S = 0.524$ and $E = 10$. We achieve an overall test accuracy of 36% on our LEAF partition.

**FEMNIST**: We take 60 clients per round. We take $E = 10, \eta_C = 0.01, \eta_S = 1$ and $E = 30$. We achieve an overall test accuracy of 83.8% on our LEAF partition.

#### A.3.2 Federated Calibration

**Global rounds** ($T$): This determines the number of federated rounds that are used to train the calibrator. Note that this is not equivalent to a global epoch during model training, instead $T = 1$ is a single step i.e., a single federated round of client participation. In the main paper we explore how $T$ effects calibration performance. We find that most methods perform best when the calibrators have been trained for a few rounds (i.e., $15 < T < 30$) but that some methods like FedTemp also perform well in one-shot settings ($T = 1$).

**Clipping norm** ($C$): When using DP we apply a clipping norm in order to guarantee bounded sensitivity. For scaling methods, this involves clipping the model update that is sent to the server for each scaling parameter. We clip the scaling parameters to have norm $C$. We found in our experiment that $C < 1$ gave best results and fix $C = 0.5$ in our DP-FL experiments. We explore the use of temperature clipping in Appendix A.3.5 in a non-DP setting and find it can have positive

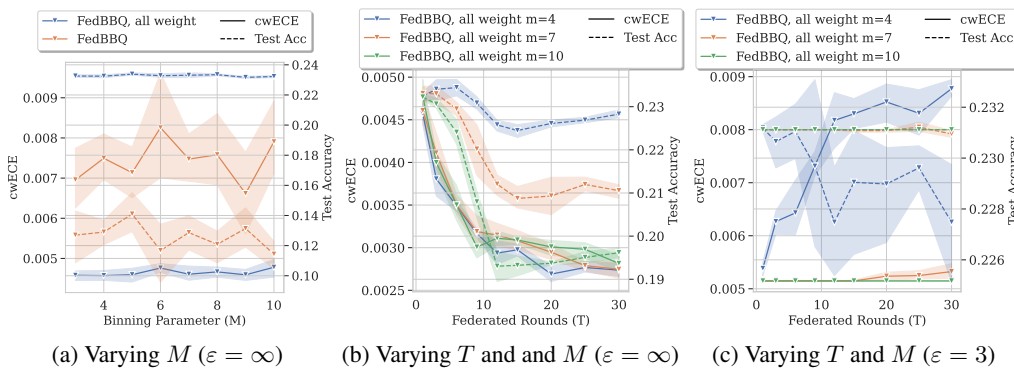

(a) Varying $M$ ($\varepsilon = \infty$)  (b) Varying $T$ and and $M$ ($\varepsilon = \infty$)  (c) Varying $T$ and $M$ ($\varepsilon = 3$)

Figure 4: Federated Calibration via FedBBQ on CIFAR100 whilst varying the bin parameter $M$. This controls the total bins for the BBQ histogram as $B = 2^M$ total bins.

improvements in mitigating label-skew. For binning methods we have two clipping norms, $[C_+, C_-]$, one for the positive histogram and one for the negative. We explore the choice of clipping norm for FedBBQ in Appendix A.3.4 and find that class-positive clipping norm $C_+$ should be chosen to be relatively small for best results.

**FedBBQ Binning Parameter** ($M$): This controls the size of the histogram created in FedBBQ resulting in total size $2^M$. In our experiments we fix this to $M = 7$. In Appendix A.3.3 we explore values of $M$ and find, although not optimal, it gives consistently good results across all settings.

**Privacy budget** ($\epsilon$): This is the differential privacy budget. We calibrate the noise needed via zCDP accounting (Bun & Steinke, 2016) see Appendix B.1 for more details.

### A.3.3 VARYING FEDBBQ BINNING PARAMETER ($M$)

In Section 4.1 we instantiate BBQ via Algorithm 1. We instantiate the total number of bins in BBQ via a parameter $M$ which, for each class, creates two histogram over class positive and negative samples with a total number of bins equal to $2^M$. Further histograms are created by merging neighboring bins to create multiple binning calibrators with bin sizes in the range of $\{2, 4, \cdots, 2^{M-1}, 2^M\}$ which are then averaged via BBQ. In our main experiments we use a fixed choice of $M = 7$. In Figure 4, we study the effect of $M$ on both the cwECE and test accuracy after calibration with FedBBQ, and demonstrate that it does not have a significant impact on the results.

In Figure 4a we vary the binning parameter $M$ with $T = 1$ rounds of BBQ in a non-DP setting with heterogeneity parameter $\beta = 0.1$ on CIFAR 100. We plot both the cwECE and the test accuracy and compare our FedBBQ all weight method against naive FedBBQ. We observe that our weighting approach has much less variance across all $M$ values with consistent test accuracy and cwECE which outperforms naive BBQ. Our choice of $M = 7$ in our experiments is justified here when $T = 1$.

In Figure 4b we vary the number of federated rounds $T$ alongside the binning parameter $M \in \{4, 7, 10\}$. Here we observe that when $T > 1$ the choice of $M$ is more important to the overall performance of FedBBQ. We observe that setting $M$ too large ($M = 10$) results in poor test accuracy for large $T$. In Figure 4c we plot the same experiment but under DP with a privacy budget of $\varepsilon = 3$. Here we observe that while $M = 4$ achieves the best balance of test accuracy and cwECE in the non-DP setting (Figure 4b) it does not perform well in the DP setting (Figure 4c). This further justifies our choice of $M = 7$ as, while not optimal across all settings, has consistently good utility (i.e., good balance of cwECE and test accuracy). We note that finding ways to adaptively select this parameter on private data in a federated way remains future work.

### A.3.4 CLIPPING NORMS: FEDBBQ

In Figure 8c we investigate the performance of binning over a variety of clipping norm choices to understand why it performs poorly compared to scaling in the user-level DP setting. We plot the average test accuracy and cwECE after calibration for clipping norms of the form $[C_+, C_-]$ where $C_+$

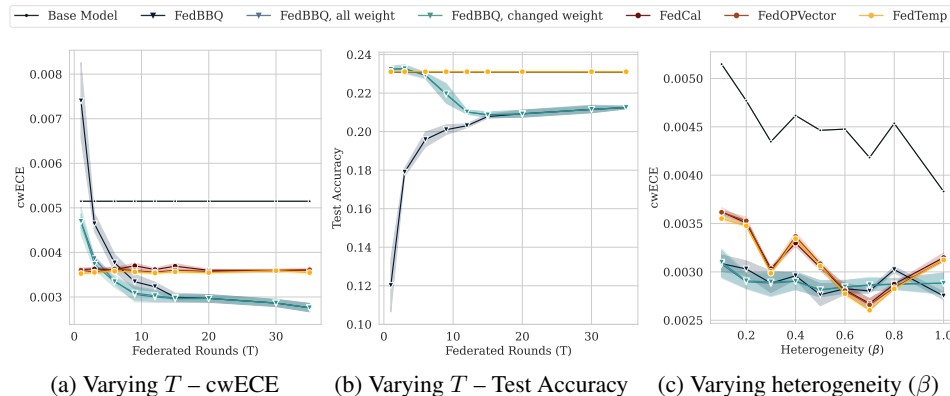

(a) Varying $T$ – cwECE     (b) Varying $T$ – Test Accuracy     (c) Varying heterogeneity ($\beta$)

Figure 5: FL Calibration on CIFAR100 (Simple CNN), $\beta = 0.1$ unless otherwise stated.

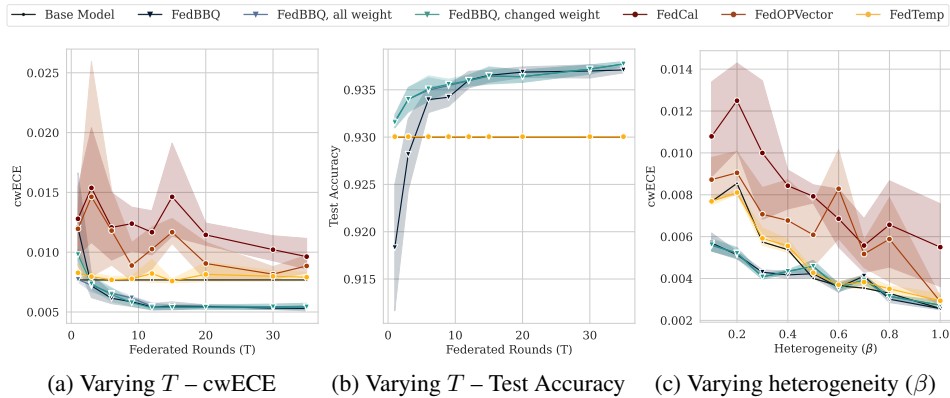

(a) Varying $T$ – cwECE     (b) Varying $T$ – Test Accuracy     (c) Varying heterogeneity ($\beta$)

Figure 6: FL Calibration on MNIST (Simple CNN), $\beta = 0.1$ unless otherwise stated.

is the clipping norm for class positive histograms and $C_-$ is the clipping norm for negative histograms. We color points by their respective positive clipping norm $C_+$. We clearly find the clipping norm has a significant impact on the resulting cwECE and test accuracy of the calibrator, and that the biggest impact is from the choice of norm for positive class histograms. This should be chosen relatively small to maintain both the low cwECE and high test accuracy.

### A.3.5   CLIPPING NORMS: FEDTEMP WITH TEMPERATURE CLIPPING

We noted in Table 2 and Table 3 that FedTemp with DP often outperformed the non-DP variant, particularly on CIFAR10. In Figure 11 we plot FedTemp on CIFAR10 (Simple CNN, $\beta = 0.1$) varying the federated calibration rounds $T$ and the clipping norm $(C)$ where $C = -1$ is no clipping. When $5 < T < 30$ we find clipping the clients temperature parameters before performing FedAvg helps improve the cwECE. This is because it prevents a single client from changing the global temperature parameter too much by their local skew. We note that clipping is not effective in a one-shot setting $(T = 1)$ or when $T$ is large. This is because when $T$ is large, the effects of local-skew are averaged out over enough steps. When $T = 1$, the initial temperature parameter is likely far from the global optimal and so clipping will actually worsen results.

### A.4   FURTHER EXPERIMENTS

### A.4.1   FEDERATED CALIBRATION

In Figure 5 and 6 we present FL calibration plots for CIFAR100 and MNIST respectively. In this setting we vary the federated rounds $(T)$ and heterogeneity $(\beta)$ and study both the classwise-ECE and test accuracy after federated calibration. We find results consistent with our conclusions in the

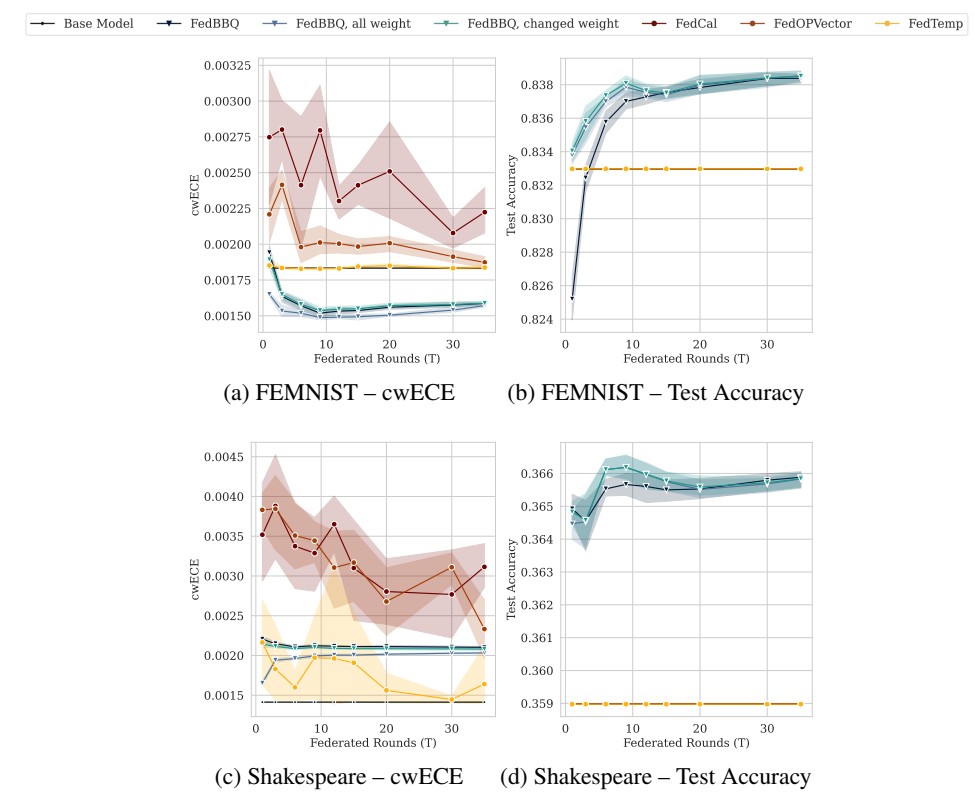

(a) FEMNIST – cwECE      (b) FEMNIST – Test Accuracy

(c) Shakespeare – cwECE      (d) Shakespeare – Test Accuracy

Figure 7: FL Calibration on LEAF – Varying $T$

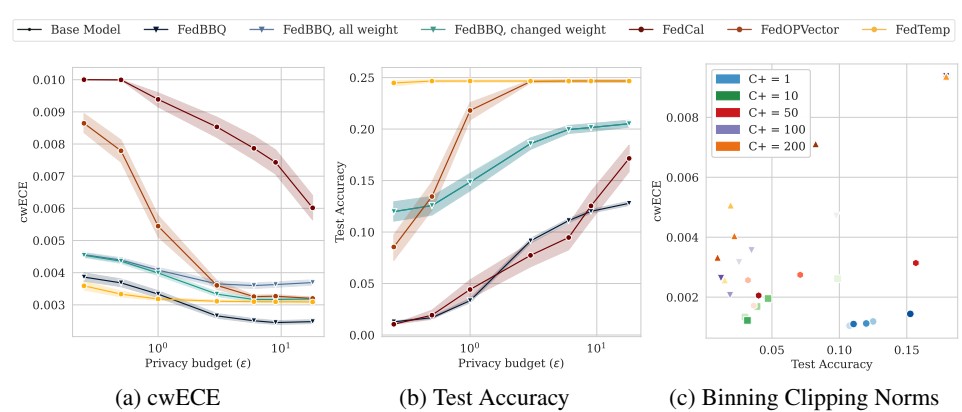

(a) cwECE      (b) Test Accuracy      (c) Binning Clipping Norms

Figure 8: DP-FL Calibration - Varying $\varepsilon$ on CIFAR100 (Simple CNN, $\beta = 0.1$)

main paper on CIFAR10. That is, scaling methods perform well in calibration over a few rounds, but the binning methods achieve best cwECE after training for multiple rounds ($T \geq 30$) as evident in Figure 5a and Figure 6a. Similarly, when we vary heterogeneity ($\beta$) we observe that when $\beta$ is small (high heterogeneity) it results in higher cwECE than in the IID setting. This trend is more apparent on MNIST (Figure 6c) than on CIFAR100 where achievable cwECE is fairly constant as heterogeneity is varied (Figure 5c). This is likely because the overall model accuracy on CIFAR100 does not change significantly as heterogeneity ($\beta$) is varied (as seen in Table 7).

In Figure 7 we present further plots on LEAF datasets. Here we only present experiments varying the number of calibration rounds ($T$) as the LEAF datasets have a fixed partitioning and so cannot vary heterogeneity ($\beta$). For FEMNIST, we observe consistent results as in previous datasets e.g., that

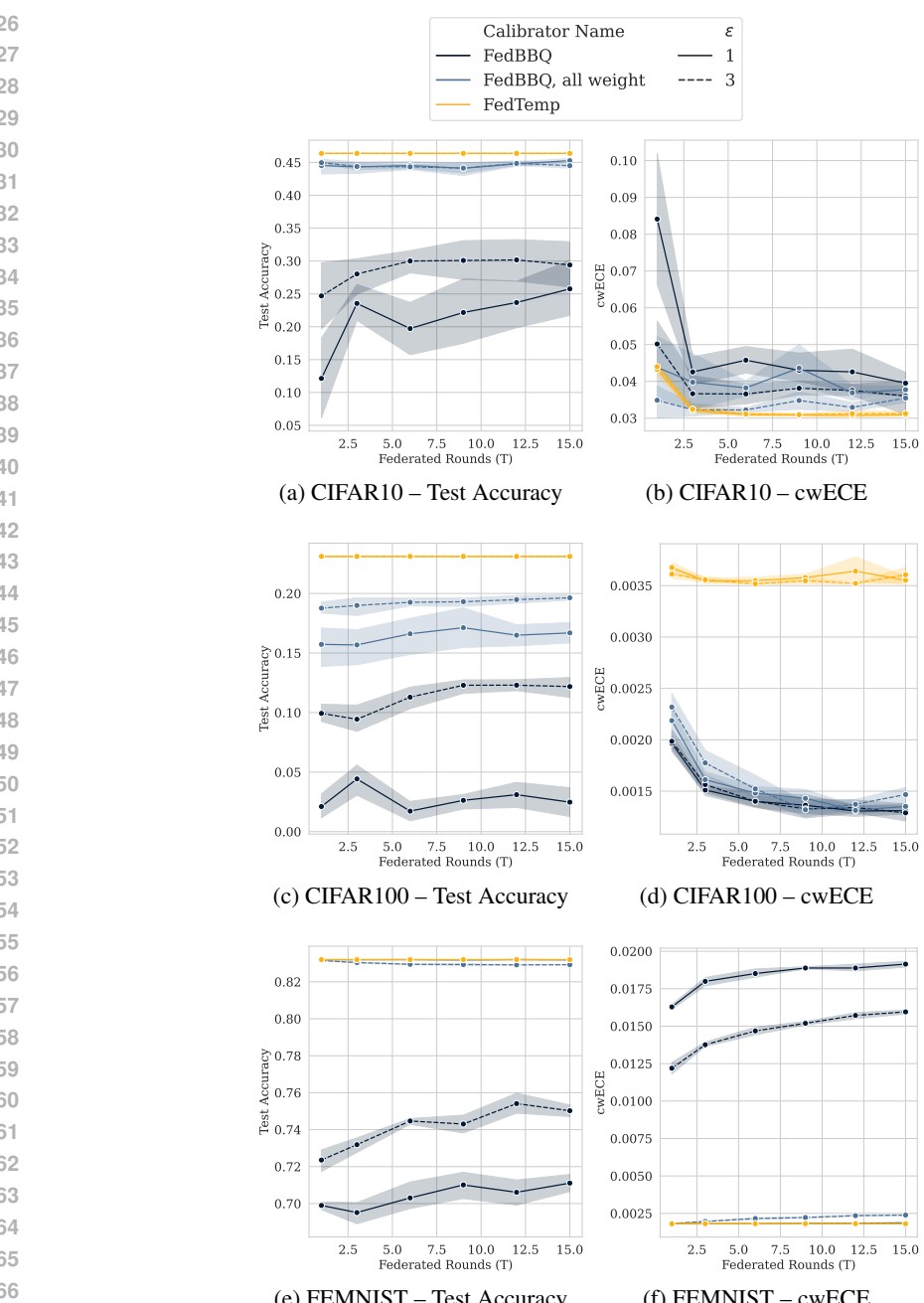

Figure 9: DP-FL Calibration, Varying $T$ and $\varepsilon$, $\beta = 0.1$ except FEMNIST which uses LEAF.

binning outperforms scaling over multiple rounds (Figure 7a). For Shakespeare, as seen in the main paper, federated calibration is difficult and no methods improve over the base model.

### A.4.2 DP-FL CALIBRATION

**Replication of Figure 3** In Figure 8 we present DP-FL calibration results for CIFAR100. Here we vary $\varepsilon$ whilst studying both cwECE and test accuracy. Results here are consistent with our conclusions made in Section 5 for DP-FL on CIFAR10, which is that, under strict DP (small $\varepsilon$), FedTemp performs best. However, CIFAR100 shows a clearer trend that as $\varepsilon$ grows large, the FedBBQ methods perform best, highlighting that FedBBQ only works well in DP settings with little noise.

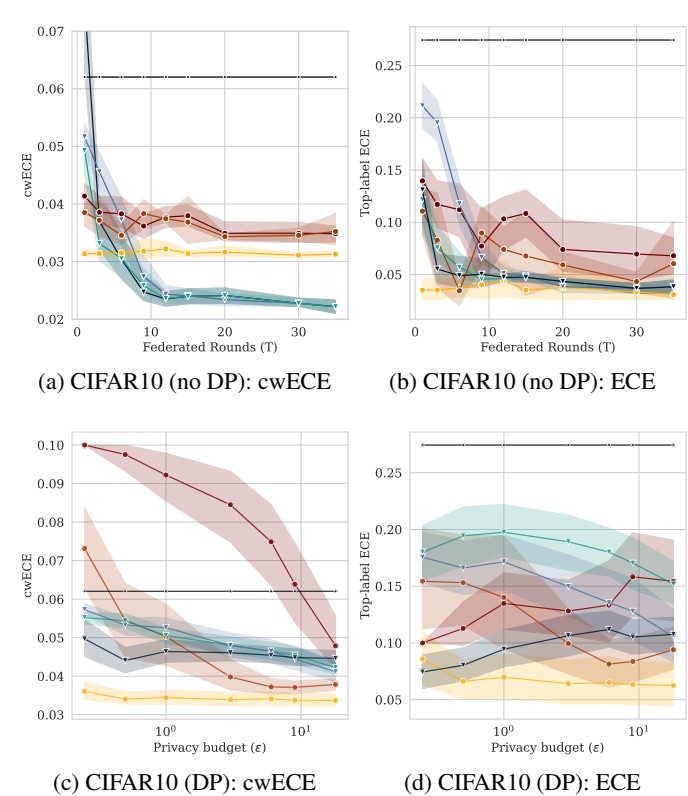

(a) CIFAR10 (no DP): cwECE      (b) CIFAR10 (no DP): ECE

(c) CIFAR10 (DP): cwECE      (d) CIFAR10 (DP): ECE

Figure 10: cwECE vs. ECE on CIFAR10 (Simple CNN, $\beta = 0.1$)

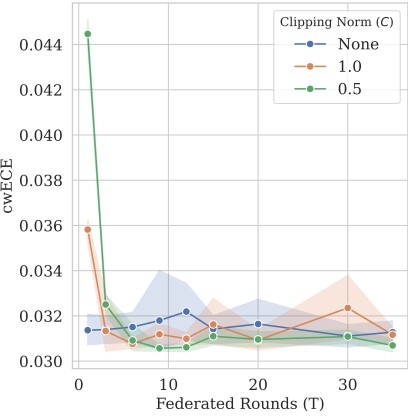

Figure 11: FedTemp on CIFAR10 (Simple CNN, $\beta = 0.1$) - Varying clipping norm $C$

**Varying $T$ and $\varepsilon$.** In Figure 9 we vary both the number of rounds ($T$) and the privacy budget ($\varepsilon$) for CIFAR10, CIFAR100 and FEMNIST. We study $\varepsilon = 1, 3$. We clearly observe that our weighting method helps preserve the test accuracy of binning calibrators, especially under strict DP ($\varepsilon = 1$). For example, on CIFAR10 (Figure 9a), the only method to significantly lose test accuracy after federated calibration are versions of FedBBQ without any weighting scheme. This is consistent on CIFAR100 and FEMNIST. For cwECE, we find FedTemp has consistent performance as both $T$ and $\varepsilon$ is varied. However, as discussed in Section 5, as the binning methods are more sensitive to noise, their performance improves (often significantly) as both $T$ and $\varepsilon$ increases.

Table 10: Simulated communication overhead of local participants who participate in federated calibration on CIFAR10 (ResNet18). The parameter $c$ is the total number of classes, $N$ is the total number of hidden neurons in the FedCal MLP scaler and $M$ is the binning parameter in BBQ.

| Calibrator | Communication Complexity | Send Size (per-round) | Receive Size (per-round) | Model Size (received once) |
|---|---|---|---|---|
| FedTemp | $O(1)$ | 0.01kb | 0.01kb | 43678.29kb |
| FedOPVector | $O(c)$ | 0.16kb | 0.16kb | 43678.29kb |
| FedCal | $O(N^2 + c^2)$ | 43.08kb | 43.08kb | 43678.29kb |
| FedBBQ | $O(2^M c)$ | 20.0kb | 0.0kb | 43678.29kb |
| FedBBQ, all weight | $O(2^M c)$ | 20.0kb | 0.0kb | 43678.29kb |

Table 11: Local client per-round computation time for federated calibration on CIFAR10 (ResNet18). Time is measured in seconds.

| Calibrator | Client Time (Mean) | Client Time (Min) | Client Time (Max) |
|---|---|---|---|
| FedBin | 0.005s | 0.003s | 0.008s |
| FedBBQ | 0.032s | 0.030s | 0.036s |
| FedBBQ, all weight | 0.039s | 0.031s | 0.049s |
| FedTemp | 0.111s | 0.099s | 0.124s |
| FedOPVector | 0.017s | 0.013s | 0.026s |
| FedCal | 0.008s | 0.006s | 0.015s |

## A.5 TOP-LABEL ECE VS. CLASSWISE ECE

In Figure 10, we present the cwECE results shown for CIFAR10 (Simple CNN, $\beta = 0.1$) in the main paper where Figure 2a is the same as Figure 10a and Figure 3a is the same as 10c. We present the equivalent figures for the (top-label) ECE metric. These demonstrate why we focus only on cwECE in our work. First, for the FL (no DP) setting in Figures 10a (cwECE) and 10b (ECE) we see that training for multiple rounds results in both binning and scaling methods achieving similar ECE scores yet the cwECE results are very different, showing a gap between scaling and binning approaches. This highlights why we focus on cwECE as it tells a more complete picture for the federated setting. For the DP-FL setting in Figures 10c (cwECE) and 10d (ECE) we find the ECE results are more consistent with the cwECE conclusions e.g., that scaling methods are preferred to binning approaches under DP.

## A.6 COMPUTATION AND COMMUNICATION OVERHEAD

In this section we present benchmarks for the computation and communication overhead of the federated calibration methods we consider in this work.

For communication overhead, we present the per-round communication cost of a client participating in federated calibration across a range of methods on CIFAR100 in Table 10. We do this in a simulated setting where we do not measure the communication overhead of using secure-aggregation protocols. Instead, these benchmarks are the raw communication cost of sending the calibration model parameters to and from the local clients. Observe the following:

- FedTemp and FedOPVector require sending 1 parameter (the temperature) and $2c$ parameters (the classwise scaling parameters) respectively. This results in very little per-round communication which is dwarfed by the cost of having to receive the federated classifier model from the server.

- FedBBQ and our FedBBQ variation with a weighting scheme has identical communication cost which requires sending a histogram of size $2^M$ for each positive and negative class resulting in $2 \cdot (2^M c)$ total communication. This is more costly than scaling approaches but better than FedCal as it does not need to receive any calibration model parameters from the server since FedBBQ just requires local histograms to be computed and sent to the server.

- FedCal requires sending a small neural network to and from the clients. We use the same architecture as was proposed in the original paper (Peng et al., 2024). This has 2 hidden

layers with 64 neurons but crucially scales with the number of classes $c$ (as the input and outputs of the MLP). This method has the largest overhead as it requires receiving and sending this MLP back to the server at each round.

In Table 11 we present the average, min and max computation time required by local clients when they participate in a single round of federated calibration. Observe that all methods are lightweight requiring at most 0.1 seconds of local compute time. We observe that FedBBQ is more costly than FedBin as it requires computing larger histograms but this is still a lightweight procedure. For scaling, we observe that FedCal has the fastest training time compared to temperature and vector scaling. This difference arises because we follow the FedCal paper default setting of only 3 training epochs trained via SGD, while our implementations of FedTemp and FedOPVector involve solving a convex optimization problem where we set a relatively large number of maximum iterations (50). Nevertheless, none of the methods introduce prohibitive computational costs, so this is not a limiting factor for federated calibration. Instead, the key consideration lies in selecting methods that achieve best calibration error whilst minimizing communication overhead as discussed above.

# B  ALGORITHM DETAILS

## B.1  DIFFERENTIAL PRIVACY

In this section, we present and prove the privacy guarantees of our federated calibration approaches under user-level DP. For completeness, we provide additional definitions and results, starting with the definition of $(\varepsilon, \delta)$-Differential Privacy.

**Definition B.1** (Differential Privacy (Dwork et al., 2014)). *A randomised algorithm $\mathcal{M} : \mathcal{D} \to \mathcal{R}$ satisfies $(\varepsilon, \delta)$-differential privacy if for any two adjacent datasets $D, D' \in \mathcal{D}$ and any subset of outputs $S \subseteq \mathcal{R}$,*

$$\mathbb{P}(\mathcal{M}(D) \in S) \le e^\varepsilon \mathbb{P}(\mathcal{M}(D') \in S) + \delta.$$

We will first provide guarantees with the more convenient $\rho$-zCDP formulation.

**Definition B.2** ($\rho$-zCDP). *A mechanism $\mathcal{M}$ is $\rho$-zCDP if for any two neighbouring datasets $D, D'$ and all $\alpha \in (1, \infty)$ we have $D_\alpha(\mathcal{M}(D)|\mathcal{M}(D') \le \rho \cdot \alpha$, where $D_\alpha$ is Renyi divergence of order $\alpha$.*

It is common to then translate this guarantee to the more interpretable $(\varepsilon, \delta)$-DP via the following lemma.

**Lemma B.3** (zCDP to DP (Canonne et al., 2020)). *If a mechanism $\mathcal{M}$ satisfies $\rho$-zCDP then it satisfies $(\varepsilon, \delta)$-DP for all $\varepsilon > 0$ with*

$$\delta = \min_{\alpha > 1} \frac{\exp((\alpha - 1)(\alpha\rho - \varepsilon))}{\alpha - 1} \left(1 - \frac{1}{\alpha}\right)^\alpha$$

In this work we are concerned with *user-level* differential privacy. In order to provide user-level DP guarantees we must bound the contribution of any one user. The following notion of (user-level) sensitivity captures this.

**Definition B.4** ($L_2$ Sensitivity). *Let $f : \mathcal{D} \to \mathbb{R}^d$ be a function over a dataset. The $L_2$ sensitivity of $f$, denoted $\Delta_2(f)$, is defined as $\Delta_2(f) := \max_{D \sim D'} \|f(D) - f(D')\|_2$, where $D \sim D'$ represents the user-level relation between datasets i.e., $D'$ is formed from the addition or removal of an entire user's data from $D$.*

A standard mechanism for guaranteeing differential privacy guarantees on the numerical outputs of an algorithm is through the use of the Gaussian mechanism.

**Definition B.5** (Gaussian Mechanism, GM). *Let $f : \mathcal{D} \to \mathbb{R}^d$, the Gaussian mechanism is defined as $GM(f) = f(D) + \Delta_2(f) \cdot \mathcal{N}(0, \sigma^2 I_d)$. The GM satisfies $\frac{1}{2\sigma^2}$-zCDP.*

As the Gaussian mechanism is invoked at each federated round, we would also like to compose these privacy guarantees.

**Lemma B.6** (zCDP composition (Bun & Steinke, 2016)). *If a mechanism $\mathcal{M}$ satisfies $\rho_1$-zCDP and mechanism $\mathcal{M}'$ satisfies $\rho_2$-zCDP, then the composition $M^*(x) := M'(M(x))$ satisfies $(\rho_1 + \rho_2)$-zCDP*

We assume a trusted honest-but-curious server adds Gaussian noise to the quantities that have been aggregated via secure aggregation as outlined in the threat model at the end of Section 3. For scaling, noise is added to the (clipped) parameters of the scaling model whilst for binning noise is added to the (clipped) histograms. For both scaling and binning, the training of the calibrator is thus the composition of the Gaussian mechanism over multiple federated rounds. We formalise this in Lemma B.7.

**Lemma B.7** (Noise calibration for $(\varepsilon, \delta)$-DP federated calibration). *For any number of federated calibration rounds $T$, federated scaling and binning approaches satisfy $(\varepsilon, \delta)$-DP, by computing $\rho$ according to Lemma B.3 and setting*

$$\sigma = \begin{cases} C\sqrt{\frac{T}{2 \cdot \rho}}, & \text{Scaling} \\ C\sqrt{\frac{2cT}{2 \cdot \rho}}, & \text{Binning} \end{cases}$$

*Proof.* As the server adds noise to the quantities received from secure-aggregation, it suffices to apply Lemma B.6 with Definition B.5. To find $\sigma$, we simply need to count the number of times the Gaussian mechanism is used. For scaling approaches, we train via DP-FedAvg which has clients send a clipped update with norm $C$ at each round. This is performed over $T$ federated rounds and hence the noise scales proportional to $T$. For binning, the privacy guarantees are the same as above except each user sends two (clipped) histograms over their datasets, one for positive and one for negative examples. For the case of multi-class classification with $c > 2$, this happens for each class. Hence users send $2c$ histograms per federated round and so the noise scales proportional to $2cT$. $\qquad\square$

For settings where the client participation rate $p < 1$ we can use subsampling amplification via Renyi Differential Privacy (RDP) to get improved composition results, see Mironov (2017) for more information.

### B.2 FEDERATED AVERAGING

We train all of our models and our scaling calibrators via variations of (DP)-FedAvg. At step $t$ of FedAvg training, we subsample clients from the population with probability $p$ and have them train their local model with current global weights $\boldsymbol{w}^t$ via local SGD for a number of epochs, producing local weights $\boldsymbol{w}_k^t$. Clients calculate a model update of the form $u_k := \boldsymbol{w}^t - \boldsymbol{w}_k^t$ which acts a pseudo-gradient. The server updates the global model of the form $\boldsymbol{w}^{t+1} := \boldsymbol{w}^t - \eta_S \cdot \frac{1}{p \cdot K} \sum_k (\boldsymbol{w}^t - \boldsymbol{w}_k^t) = \boldsymbol{w}_k^t - \eta_S \cdot \frac{1}{p \cdot K} \sum_k u_k$, where $\eta_S$ is a server learning rate. In the context of DP-FedAvg, the individual model updates $u_k$ are clipped to have norm $C$ and aggregated by the server via secure-aggregation. Denoting $\bar{u}_k = \text{clip}(u_k, C)$ the server computes the final noisy update of the form $\tilde{u} := \sum_k \bar{u}_k + N(0, C\sigma^2 I_d)$

### B.3 HISTOGRAM BINNING

One of the simplest calibration methods is histogram binning (Zadrozny & Elkan, 2002). In a binary classification setting, given the output confidence $\hat{p}_i \in [0, 1]$, class-label $y_i \in \{0, 1\}$ and a fixed total number of bins $M$, the output confidences of the model are partitioned using fixed-width bins of the form $B_m = [\frac{m-1}{M}, \frac{m}{M}]$ for $m \in \{1, \cdots, M\}$. The calibrator $g(\hat{p})$ is then formed as

$$g(\hat{p}) = \sum_m \mathbf{1}\{\hat{p}_i \in B_m\} \frac{P(m)}{P(m) + N(m)}$$

Where the entries of the class-positive histogram $P$ and class-negative histogram $N$ is defined as

$$P(m) := \sum_i \mathbf{1}\{y_i = 1 \land \hat{p}_i \in B_m\}$$

$$N(m) := \sum_i \mathbf{1}\{y_i = 0 \land \hat{p}_i \in B_m\}$$

To extend this to a multiclass setting, we can learn $c$ one-vs-all calibrator models $g_1, \cdots g_c$ for each class. For a particular example $i$ and class $j$, with predicted class-confidence $\hat{p}_{i,j}$, the class-positive and negative histograms $P_j(m)$ and $N_j(m)$ are computed as

$$P_j(m) := \sum_i \mathbf{1}\{y_i = j \wedge \hat{p}_{i,j} \in B_m\}$$

$$N_j(m) := \sum_i \mathbf{1}\{y_i \neq j \ \wedge \hat{p}_{i,j} \in B_m\}$$

The final calibrator $g(\cdot)$ is formed from the normalized one-vs-all calibrators as

$$g(\hat{\mathbf{p}_i}) := (g_1(\hat{p}_{i,1}), \cdots g_c(\hat{p}_{i,c}))/ \sum_j g_j(\hat{p}_{i,j})$$

### B.4 Bayesian Binning Quantiles (BBQ)

The BBQ approach of Naeini et al. (2015) extends histogram binning to consider multiple binning schemes at once. Given a histogram calibrator with $B$ total bins, BBQ assigns a score based on the Gamma function ($\Gamma(\cdot)$) of the form:

$$\text{Score}(M) := \prod_{b=1}^{B} \frac{\Gamma(\frac{N'}{B})}{\Gamma(N_b + \frac{N'}{B})} \frac{\Gamma(m_b + \alpha_b)}{\Gamma(\alpha_b)} \frac{\Gamma(n_b + \beta_b)}{\Gamma(\beta_b)}$$

where $N_b$ is the total number of samples in bin $b$, $n_b$ is the number of negative class samples in bin $b$, $m_b$ the number of positive class samples and $\alpha_b := \frac{2}{B} p_b, \beta_b := \frac{2}{B}(1 - p_b)$ where $p_b$ is the midpoint of the interval defining bin $b$. Given a set of binning calibrators $M_1, \ldots M_n$ with associated scores $s_1, \ldots, s_n$, the final BBQ calibrator is formed from the weighted average $M(p) := \frac{\sum_i s_i \cdot M_i(p)}{\sum_i s_i}$. In the multi-class setting we consider a one-vs-all setting where we train a BBQ calibrator for each class. In our federated setting, and to simplify things under DP, we consider a histogram calibrator with $2^M$ total bins. We have clients compute positive and negative histograms over each class for this binning scheme. Then to utilize BBQ, we merge the $2^M$ bins of each histogram in powers of 2 to create multiple histogram calibrators with total bin counts in the range $\{2, 4, \ldots, 2^{M-1}, 2^M\}$ and assign the BBQ scores defined above. This defines our FedBBQ protocol.

### B.5 Weighting Scheme for Binning Methods

In the non-DP setting we apply a classwise weighting scheme with $\alpha_j = \text{clip}(\frac{\tilde{N}_j}{N_j}, 1)$ where $\tilde{N}_j$ is the number of aggregated class $j$ examples and $N_j$ is the total number of class $j$ examples. Note that this weighting scheme requires no additional rounds of communication from clients as it is simply a post-processing step the server performs (summing a histogram) on the aggregated histogram received from clients. In the DP setting, knowledge about $N_j$ is private. We can still estimate $\tilde{N}_j$ under DP by summing all the counts from the (noisy) positive histogram for class $j$. We replace $N_j$ with the expected error of measuring the histogram under Gaussian noise, $\tilde{\alpha}_j := \text{clip}(\tilde{N}_j/\sqrt{2/\pi}\sigma|P_j|, 1)$. In other words, if the signal-to-noise ratio of the histogram is low then we place less weight on the binning prediction.

### B.6 Order-preserving Training for Scaling Methods

The method of Rahimi et al. (2020) allows training a calibrator $g$ to be *order-preserving*.

**Definition B.8** (Order-preserving). *We say a calibrator $g$ is order-preserving for any $\mathbf{p}$ if both $\mathbf{p}$ and $g(\mathbf{p})$ share the same ranking i.e., for all $i, j \in [c]$ we have $\mathbf{p}_i \geq \mathbf{p_j}$ if and only if $g_i(\mathbf{p}) \geq g_j(\mathbf{p})$*

This guarantees that the top-label prediction (i.e., top-1 accuracy) will not be changed after calibration. Rahimi et al. show that a necessary and sufficient condition for $g$ to be order-preserving is if $g$ is of the form

$$g(\mathbf{p}) := S(\mathbf{p})^{-1} U \mathbf{w}(\mathbf{p}) \tag{1}$$

with $U$ being an upper-triangular matrix of ones, $S$ being a sorting permutation and $\mathbf{w}$ a function that satisfies

- $\mathbf{w}_i(\mathbf{p}) = 0$ if $\mathbf{y}_i = \mathbf{y}_{i+1}$ and $i < n$
- $\mathbf{w}_i(\mathbf{p}) > 0$ if $\mathbf{y}_i > \mathbf{y}_{i+1}$ and $i < n$

where $\mathbf{y} := S(\mathbf{p})\mathbf{p}$ is the sorted version of the class-probabilities $\mathbf{p}$ (see Theorem 1 in Rahimi et al. (2020)). Rahimi et al. further show that for any scaling calibrator $\mathbf{m}(\mathbf{p})$, this can be achieved by setting $\mathbf{w}(\mathbf{p}) := |\mathbf{y}_i - \mathbf{y}_{i+1}|\mathbf{m}(\mathbf{p})$ and plugging $\mathbf{w}(\mathbf{p})$ into (1) to obtain the final order-preserving calibrator $g$.

