# OpenReview forum: "Private Federated Multiclass Post-hoc Calibration"
_ICLR.cc/2026/Conference — Submitted to ICLR 2026_

### Official Review · Reviewer_6xAV · 2025-10-26

**Soundness:** 2
**Presentation:** 2
**Contribution:** 2
**Rating:** 2
**Confidence:** 5

**Summary:**

This paper studies post-hoc calibration of multiclass classifiers in federated learning, with a focus on non-IID client data and user-level differential privacy. The authors adapt two common calibration strategies, histogram binning and temperature scaling, to the federated setting. For binning, they propose FedBBQ, which extends Bayesian Binning into Quantiles to FL and introduces a weighting scheme intended to handle client heterogeneity. For scaling, they propose a federated version of temperature scaling with an order-preserving training procedure that aims to avoid accuracy loss when clients are highly different. They also provide differentially private variants by clipping client updates and adding calibrated noise. The experiments span seven datasets. The results suggest that FedBBQ gives the best calibration in non-private FL with strong heterogeneity, while FedTemp is the most stable and effective approach when differential privacy is enforced.

**Strengths:**

1. The paper evaluates its methods on a range of datasets in both standard federated learning and differentially private federated learning settings.

2. It proposes several calibration strategies that target different degrees of client heterogeneity and different privacy budgets, which makes the approach more practical and more adaptable to real deployment scenarios.

**Weaknesses:**

1. The paper still lacks some key formal definitions in the main text. For example, the Bayesian binning (BBQ) procedure, the exact weighting scheme used by the server to stabilize FedBBQ under heterogeneity, and the “order-preserving” training used in scaling are only described at a high level or deferred to the appendix. Without these details in the core paper, it is hard for the reader to fully understand or reproduce the proposed methods.

2. The motivation and analysis behind the main technical ideas could be stronger. The paper introduces weighted binning and order-preserving scaling as fixes for heterogeneity and accuracy drop, but it does not explain in a principled way why these fixes should work or under what assumptions they are guaranteed to help. As written, they read more like heuristics that happened to improve results, rather than methods whose behavior is theoretically understood.

3. The motivation for doing post-hoc calibration in federated learning is not fully convincing. The method assumes each client can hold out a clean “calibration set” after training, and can revisit the model for an extra calibration phase. In realistic FL deployments, clients may not have enough leftover data or may not know in advance that calibration will happen. The paper does not clearly explain why calibration must be done post-hoc instead of training a calibrator jointly with the model during FL.

4. The set of baselines is limited. Most comparisons are against FedCal and simple scaling variants, plus unweighted binning. There are other relevant directions in the related work, including prior FL calibration approaches, DP-aware calibration, and non-federated multiclass calibration methods that could be adapted. The argument that prior work does not consider cwECE or user-level DP is not fully sufficient, because cwECE can still be measured after the fact and DP-SGD style noise can in principle be applied to competing methods. A broader comparison in both non-private FL and DP-FL settings would make the empirical claims more convincing.

5. There are some consistency and interpretability issues in the analysis. For example, the text claims that the proposed fixes prevent accuracy degradation under heterogeneity, but Table 2 still marks cases where FedBBQ causes more than 1% drop in test accuracy. The paper also argues that cwECE is the “right” metric for multiclass FL, mainly because it separates classes better than top-label ECE, but this is more a sensitivity argument than a justification that cwECE reflects real calibration quality for downstream decisions. These points should be clarified.

6. The proposed FedBBQ method appears highly sensitive to hyperparameters such as the number of bins, the way multiple histograms are merged, the weighting factor αj, the clipping norms for DP, and the number of calibration rounds T. The paper does not provide clear guidance on how to choose these values, which makes it hard to know how robust the method is across new datasets

7. The paper claims that the proposed approaches are lightweight in terms of computation and communication, but the main text does not quantify this cost. For example, the overhead of sending per-class histograms in FedBBQ, or repeatedly sending scaling parameters under DP noise, is not analyzed in detail. Since FL and DP-FL are often bottlenecked by communication and client resource limits, these costs should be made explicit in the main results rather than only mentioned qualitatively.

**Questions:**

1. The paper introduces a weighting scheme (for binning) and an order-preserving training strategy (for scaling), and claims these help under client heterogeneity. Could you provide more intuition for how these components work? In particular, how exactly do they reduce the negative impact of heterogeneous client distributions, and why do they help avoid accuracy drop during calibration?

2. FedTemp shows surprisingly strong performance under high heterogeneity and when differential privacy is applied. Can you explain why simple temperature scaling is so effective in that regime? Is there an intuitive reason that temperature scaling is more robust than the other approaches when data is both non-iid and noisy due to DP?

---

> ### Author Response · Authors · 2025-11-20
> **Rebuttal 1/X**
>
> We would like to thank the reviewer for their comments.
>
> > The paper still lacks some key formal definitions in the main text…. the Bayesian binning (BBQ) procedure, the exact weighting scheme used by the server to stabilize FedBBQ under heterogeneity, and the “order-preserving” training used in scaling…
>
> Reproduction is possible from the provided algorithms, formulas and the open-source code. We include detailed hyperparameter settings for 1) the training of the base FL model and 2) the calibration process. All of the details you have stated are included in the main paper or appendix as follows:
> 1. The weighting scheme is given explicitly in the main text (Section 4.1, L305). Extra details are included in Appendix B.5.
> 2. FedBBQ is described in the main paper with the original BBQ algorithm fully defined in Appendix B.4.
> 3. Histogram binning and its multiclass extensions are explained in Section 4.1 and explicitly listed out in Appendix B.3
> 4. Order-preserving training is formally defined in Appendix B.6
>
> If you believe some further explanation is needed to ensure clarity and reproducibility, we will be happy to add it.
>
> > The motivation for doing post-hoc calibration in federated learning is not fully convincing. The method assumes each client can hold out a clean “calibration set” after training, and can revisit the model for an extra calibration phase. In realistic FL deployments, clients may not have enough leftover data or may not know in advance that calibration will happen.
>
> In our experiments we assume clients hold out a clean calibration set for simplicity, but this is not a strict requirement in practice. In realistic cross-device FL with many thousands or millions of users (representing the real-world deployments that have been documented by Google, Meta, Apple etc.), many clients may never have been selected during training and can devote all of their local data for calibration. Post-hoc calibration also captures a broader range of scenarios where the model need not be trained in a federated manner but still needs to be calibrated across federated clients. Moreover, retraining from scratch with train-time calibration is often prohibitively expensive in practical FL pipelines, and, as our new experiments show (see below) not guaranteed to work. In contrast, post-hoc calibration is a lightweight, effective option that can be used whenever calibration needs arise.
>
> For experiments against train-time methods, please see our response in the second comment which addresses this.
>
> > There are some consistency and interpretability issues in the analysis... the text claims that the proposed fixes prevent accuracy degradation under heterogeneity, but Table 2 still marks cases where FedBBQ causes more than 1% drop in test accuracy. The paper also argues that cwECE is the “right” metric for multiclass FL, mainly because it separates classes better than top-label ECE, but this is more a sensitivity argument than a justification that cwECE reflects real calibration quality for downstream decisions.
>
> We clarify that Table 2 flags methods with >1% accuracy drop; the stars appear on the naive (unweighted) FedBBQ, not on our weighted FedBBQ variant (“FedBBQ, all weight”),  so our method does prevent accuracy degradation under heterogeneity.
>
> On the second point about cwECE, we devote Appendix A.5  to demonstrating why cwECE is more informative than top-label ECE in our multiclass FL setting, including side-by-side plots that illustrate ECE “hides” classwise differences.
>
> > The paper claims that the proposed approaches are lightweight in terms of computation and communication, but the main text does not quantify this cost. For example, the overhead of sending per-class histograms in FedBBQ, or repeatedly sending scaling parameters under DP noise, is not analyzed in detail. Since FL and DP-FL are often bottlenecked by communication and client resource limits, these costs should be made explicit in the main results rather than only mentioned qualitatively.
>
> We provide the full evidence to justify this claim about computation and communication overheads in Appendix A.6 which contains communication complexity,  per-round client communication (bytes sent/received) and local computation time for all methods. These results show that all methods are lightweight relative to training the base FL model.
>
> Based on yours and other reviewer feedback we have used the extra allotted page to move these results into the main paper in our revised version (Table 5). We hope that placing these results in the main text addresses your point clearly.

---

> ### Author Response · Authors · 2025-11-20
> **Rebuttal 2/X**
>
> > The set of baselines is limited. Most comparisons are against FedCal and simple scaling variants, plus unweighted binning. There are other relevant directions in the related work, including prior FL calibration approaches, DP-aware calibration, and non-federated multiclass calibration methods that could be adapted. The argument that prior work does not consider cwECE or user-level DP is not fully sufficient...
>
> We have observed that it is not at all straightforward to extend existing central methods under DP-SGD (or DP-FedAvg) style noise effectively without completely destroying model accuracy. We extend and propose novel methods related to binning and scaling methods which are amenable to a secure-aggregation + DP noise framework. If you know a fully specified method in the literature that you can point to, we will incorporate this into our framework.
>
> To cast light on your point, we have added experiments related to adaptations of existing train-time calibration approaches into our revised paper. Please see Section 5 and Table 5 (in the revised paper). In more detail, we adopt three train-time methods [1,2,3] that were proposed in the central setting and one train-time method designed for the federated setting [4]. We observe the following three key points:
> 1. Train-time methods are inconsistent in the federated setting. Sometimes they improve accuracy + cwECE and sometimes they make things worse than normal FedAvg.
> 2. In all cases, performing (federated) post-hoc calibration to the FedAvg model results in better accuracy + cwECE than just relying on a train-time method.
> 3. We find the best accuracy + cwECE is achieved when combining train-time with post-hoc calibrators (our methods). This aligns with the conclusions of [3] and [4] who note that train-time and post-hoc methods are complementary in the central setting.
> These conclusions only strengthen our work and our decision to focus on post-hoc calibration. We view train-time calibration as mostly orthogonal to our work, since, as our results show, they can be combined with our methods to achieve the best calibration performance.
>
> Note we only study a non-DP setting for our comparison against train-time methods as it is non-trivial to effectively translate train-time methods to the DP setting without destroying the utility of the model and this research question is out of scope for our work which focuses solely on post-hoc calibration.
>
> [1] *Marx, Charlie, Sofian Zalouk, and Stefano Ermon. "Calibration by distribution matching: Trainable kernel calibration metrics." Advances in Neural Information Processing Systems 36 (2023): 25910-25928.*
>
> [2] *Mukhoti, Jishnu, et al. "Calibrating deep neural networks using focal loss." Advances in neural information processing systems 33 (2020): 15288-15299.*
>
> [3] *Kumar, Aviral, Sunita Sarawagi, and Ujjwal Jain. "Trainable calibration measures for neural networks from kernel mean embeddings." International Conference on Machine Learning. PMLR, 2018.*
>
> [4] *Chu, Yun-Wei, et al. "Unlocking the potential of model calibration in federated learning." arXiv preprint arXiv:2409.04901 (2024).*
>
> > The proposed FedBBQ method appears highly sensitive to hyperparameters such as the number of bins, the way multiple histograms are merged, the weighting factor αj, the clipping norms for DP, and the number of calibration rounds T. The paper does not provide clear guidance on how to choose these values...
>
> Our experiments suggest the opposite: FedBBQ is not highly sensitive to hyperparameters. How we merge histograms is fixed, as they are simply aggregated (summed up). The weighting factor $\alpha_j$, by definition is fixed at each round and is not a hyperparameter we choose. The only remaining parameter specific to FedBBQ is the bin parameter $M$ which we show is robust in Appendix A.3.3. We fix this to M=7 in our experiments and do not tune it for utility and find good performance in all settings.
>
> Additionally, for other hyperparameters we argue that 1) the performance of methods is robust to the choice of parameter and 2) certain values for parameters give good performance across all settings. These results are detailed in Appendix A.3 but to summarise for T and C:
> 1. **Number of rounds (T)**:  We systematically vary T (Figure 2b) and observe that most methods (especially FedBBQ) perform best in a moderate range (roughly 15–30 rounds), while FedTemp is effective even with T=1. This parameter does not need to be tuned and setting T=30 gives good performance in the settings we studied.
> 2. **Clipping norms (C)**: For scaling methods, we vary the DP clipping norm over a small grid and find that relatively small norms improve robustness and stability, especially under label skew. This is not sensitive to the choice of $C$ as long as it is not chosen to be extremely large (i.e., > 2) For FedBBQ, we study separate clipping norms for positive and negative histograms and identify regimes where cwECE and accuracy are preserved.

---

> > ### Author Response · Authors · 2025-11-20
> > **Rebuttal 3/3**
> >
> > > FedTemp shows surprisingly strong performance under high heterogeneity and when differential privacy is applied. Can you explain why simple temperature scaling is so effective in that regime? Is there an intuitive reason that temperature scaling is more robust than the other approaches when data is both non-iid and noisy due to DP?
> >
> > This is an important point and something we comment on in L474-475 and fully address in Appendix A.3.5.
> >
> > To summarise, we find FedTemp is robust because the clipping involved in DP actually helps to reduce the impact of skew, preventing any single highly skewed client from dominating the final temperature T. Temperature scaling also only rescales logits (no bias term), so predictions cannot  change drastically under DP noise (i.e., top-k accuracy is preserved).

---

### Official Review · Reviewer_aLWK · 2025-10-30

**Soundness:** 3
**Presentation:** 3
**Contribution:** 2
**Rating:** 4
**Confidence:** 3

**Summary:**

This paper introduces two frameworks for federated post-hoc calibration in DP settings. The authors validate the effectivenss of the proposed methods on 7 datasets.

**Strengths:**

1. This paper focues on an important issue, calibration in federated settings with user-level DP, which is highly relevant for real-world application in scenarios like healthcare.

2. Generally, the paper is easy to follow.

**Weaknesses:**

1. The technical contribution of the paper appears to be somewhat limited. The proposed method primarily integrates several well-established techniques. For example, federated binning largely follows (Cormode & Markov, 2023), scaling via FedAvg is straightforward application of existing methods.

2. Some technical details aren’t well justified. For instance, in line 303, the weight $\alpha_j$ doesn’t have a clear theoretical explanation. why was this specific form chosen? The paper briefly mentions an ablation on this weight, but the comparison isn’t really explored in depth.

3. Only compares against FedCal (Peng et al., 2024) for scaling methods and missing comparisons with other recent federated calibration work.

**Questions:**

1. How do your methods scale to $K >> 100$ clients with more extreme heterogeneity?

2. The privacy analysis assumes fixed hyperparameters (M, C, etc.). In practice, how should practitioners account for the privacy cost of tuning these parameters on private data?

---

> ### Author Response · Authors · 2025-11-20
> **Rebuttal 1/2**
>
> We would like to thank the reviewer for their comments.
>
> > The technical contribution of the paper appears to be somewhat limited... For example, federated binning largely follows (Cormode & Markov, 2023), scaling via FedAvg is straightforward application...
>
> Our work goes beyond simply integrating existing techniques. Cormode & Markov (2023) provide a baseline approach to federated histogram binning under example-level DP for binary classification, whereas we design FedBBQ specifically for multiclass cwECE under user-level DP and non-IID data. This is a very different setting and much closer to the practical regime in which FL is deployed. This requires several non-trivial changes:
> 1. a merged histogram construction to keep the user-level DP cost from scaling with the number of binners.
> 2. a classwise weighting scheme that prevents accuracy collapse under strong heterogeneity (label skew)
> 3. clipping choices tailored to one-vs-all histograms in user-level DP
>
> Similarly, while FedAvg as an optimizer is straightforward, our contribution on the scaling side is to show that naively using FedAvg (via federated vector/matrix scaling) can significantly harm accuracy under heterogeneity and then to introduce an order-preserving variant (FedOPVector) which guarantees no loss in accuracy while improving cwECE.  To our knowledge, this is also the first work that provides a unified framework to study the problem of private federated post-hoc calibration in a setting which matches practical FL deployments.
>
> > Some technical details aren’t well justified. For instance, in line 303, the weight doesn’t have a clear theoretical explanation. why was this specific form chosen?
>
> The classwise weight in L303 is designed to balance the contribution of the calibrated prediction with the original model’s prediction in a way that is robust to heterogeneity, namely against label-skew.
>
> The specific form of $\alpha_j$ was chosen based on the following intuition: when the aggregated histogram of confidences for class j is estimated from many clients with a sufficient number of class j examples, we can trust the calibrator more as it will have a more accurate estimate. This is because heterogeneity most severely affects histograms when the partial participation of clients is very low. If all clients were to participate in a single round, an aggregated histogram is the same as in the central setting and so we should fully trust it (as there would be no loss in utility compared to the central setting).
>
> We have made this intuition clearer in the revised version of our paper.
>
> > Only compares against FedCal (Peng et al., 2024) for scaling methods and missing comparisons with other recent federated calibration work.
>
> Regarding baselines for scaling, our focus is on post-hoc calibration in FL, not train-time modifications. Among post-hoc federated methods, FedCal (Peng et al., 2024) is, to our knowledge, the main directly comparable baseline. Other recent FL calibration methods either (i) modify the training objective or architecture (train-time calibration), or (ii) do not handle user-level DP. If you are aware of  relevant techniques that can be applied to this problem, please do reference them so we can include them in our study.
>
> Furthermore, based on this and other reviewer comments, we have run additional experiments adapting existing train-time calibration methods into our federated setting. In our revised version, we have added this discussion and a new table (Table 4) into Section 5 of the main paper. **Please see our response to Reviewer JRNY that covers this in detail**.
>
> > How do your methods scale to clients with more extreme heterogeneity?
>
> We study how our methods scale with heterogeneity in Figure 5c. We vary the heterogeneity parameter $\beta$ where smaller $\beta$ represents more label-skew (= more difficult to perform federated calibration). The full details for how we model heterogeneity is outlined in Appendix A.2.  We find that in this setting we can achieve good calibration results under extreme heterogeneity (i.e., improve the calibration error of the base model).
>
> Note that for the majority of the experiments we fix $\beta=0.1$ which is the most extreme heterogenous setting we study. In the revised version of our paper, we have added Table 8 to the appendix which presents the average number of unique classes that clients have as we vary $\beta$. We observe when $\beta =0.1$ most clients are missing between 60-80% of classes in their local datasets which makes federated calibration very challenging.

---

> > ### Author Response · Authors · 2025-11-20
> > **Rebuttal 2/2**
> >
> > > The privacy analysis assumes fixed hyperparameters (M, C, etc.). In practice, how should practitioners account for the privacy cost of tuning these parameters on private data?
> >
> > We argue that in practice, practitioners would not need to account for the privacy cost of tuning these parameters because we show 1) the performance is robust to the choice of parameter; 2) certain values for parameters give good performance across all settings. In more detail:
> > 1. **Number of rounds (T):** We systematically vary T (Figure 2b) and observe that most methods (especially FedBBQ) perform best in a moderate range (roughly 15–30 rounds), while FedTemp is effective even with T=1. This parameter does not need to be tuned and setting T=30 gives good performance in the settings we studied.
> > 2. **Clipping norms (C):** For scaling methods, we vary the DP clipping norm over a small grid and find that relatively small norms improve robustness and stability, especially under label skew. This is not sensitive to the choice of $C$ as long as it is not chosen to be extremely large (i.e., > 2). For FedBBQ, we study separate clipping norms for positive and negative histograms and identify regimes where cwECE and accuracy are preserved.
> > 3. **BBQ bin count (M):** We vary M (Appendix A.3.3) and show that an intermediate choice (M=7, the value we use in the main experiments) is robust across both non-DP and DP-FL settings. This does not need tuning in the federated setting for good performance.

---

### Official Review · Reviewer_JRNY · 2025-10-31

**Soundness:** 3
**Presentation:** 3
**Contribution:** 3
**Rating:** 4
**Confidence:** 3

**Summary:**

The paper addresses the problem of probability calibration in FL. The motivation of the work is based on the observation that the porting centralized calibration methods to FL do not work; and local calibrations fit to local data distributions do not generalize well when aggregated reducing the accuracy of the model specially in the non-IID settings. The paper discusses two frameworks for calibration in FL - i) FedBinning which is a federated version of the histogram Binning method where clients share per class histograms and the server aggregates them; ii) FedScaling which is a federated version of the scaling method which are trained similarly to FedAvg . Both these methods can be used for multi-class settings using classwise expected calibration error (cwECE). The paper also discusses the privacy preserving versions of these two frameworks using DP.

**Strengths:**

1. The work tackles an important FL problem particularly so in real world deployment.
2. The paper is well written and easy to follow.
3. The proposed solution are easy to implement on top of existing methods and seem to have better calibration.
4. Both private and non-private versions of the methods are discussed.

**Weaknesses:**

1. The novelty in the solution stems from adapting the existing calibration methods for the FL settings with somewhat simpler changes.
2. There is little formal analysis if why these mechanisms would work under heterogeneity.

**Questions:**

1. How would the method perform under the different degrees of class imbalance across clients, for example when some classes are entirely missing from some clients?
2. How would the method compare to train time calibration methods?

---

> ### Author Response · Authors · 2025-11-20
> **Rebuttal 1/1**
>
> We would like to thank the reviewer for their comments.
>
> > The novelty in the solution stems from adapting the existing calibration methods for the FL settings with somewhat simpler changes.
>
> While our methods build on existing calibration ideas, the contributions go beyond simple adaptations. Our paper studies the first framework for multiclass calibration in the non-IID user-level, DP-FL setting. This is a setting not handled by prior work and is more aligned with practical applications of FL compared to prior research.
>
> We show that direct translation of methods (BBQ, vector scaling, matrix scaling) do not work well under realistic non-IID scenarios and propose mitigations for this. This includes FedBBQ, which is not a direct translation of histogram binning: it adds a heterogeneity-aware weighting scheme and a merged-histogram construction that is necessary for good performance under user-level DP, with a clipping mechanism tailored to one-vs-all classification.
>
> > How would the method perform under the different degrees of class imbalance across clients, for example when some classes are entirely missing from some clients?
>
> Our experiments already address these scenarios, since the label-skew that we evaluate (via varying $\beta$) creates exactly this level of class imbalance.
>
> To answer your question with concrete numbers, in the revised version of our paper, we have added a new table to the appendix (Table 8). This table presents the average number of unique classes that clients have as we vary the heterogeneity parameter $\beta$ (smaller = more skew) across MNIST, CIFAR10 and CIFAR100. We observe that when $\beta$ < 0.9 the average client is missing entire classes and that our methods are capable of good performance in this setting i.e., we improve cwECE over the base model under the presence of extreme class imbalance.
>
> > How would the method compare to train time calibration methods?
>
> Based on yours and other reviewers comments we have run additional experiments to demonstrate the limitations of incorporating existing train-time calibration methods into our federated setting. In our revised version, we have added this discussion and a new table (Table 4) into Section 5 of the main paper.
>
> In more detail, we evaluate FL adaptations of three train-time methods [1,2,3] that were proposed in the central setting and one train-time method designed for the federated setting [4]. We observe the following three key points:
> 1. Train-time methods are **highly** inconsistent in the federated setting. Sometimes they improve accuracy + cwECE and sometimes they make things worse than normal FedAvg.
> 2. In all cases, performing (federated) post-hoc calibration via our methods to the FedAvg model results in better accuracy and cwECE than just relying on a train-time method.
> 3. We find the best accuracy and cwECE is achieved when combining train-time with post-hoc calibrators (our methods). This aligns with the conclusions of [3] and [4] who note that train-time and post-hoc methods are complementary in the central setting.
>
> These conclusions further strengthen our work and our decision to focus on post-hoc calibration. We view train-time calibration as mostly orthogonal to our work, since, as our results show, they can be combined with our methods to achieve the best calibration performance.
>
> Note that we only study a non-DP setting for our comparison against train-time methods as it is non-trivial to effectively translate train-time methods to the DP setting without destroying the utility of the model and this research question is out of scope for our work which focuses solely on post-hoc calibration.
>
> [1] *Marx, Charlie, Sofian Zalouk, and Stefano Ermon. "Calibration by distribution matching: Trainable kernel calibration metrics." Advances in Neural Information Processing Systems 36 (2023): 25910-25928.*
>
> [2] *Mukhoti, Jishnu, et al. "Calibrating deep neural networks using focal loss." Advances in neural information processing systems 33 (2020): 15288-15299.*
>
> [3] *Kumar, Aviral, Sunita Sarawagi, and Ujjwal Jain. "Trainable calibration measures for neural networks from kernel mean embeddings." International Conference on Machine Learning. PMLR, 2018.*
>
> [4] *Chu, Yun-Wei, et al. "Unlocking the potential of model calibration in federated learning." arXiv preprint arXiv:2409.04901 (2024).*

---

### Official Review · Reviewer_AaLR · 2025-11-04

**Soundness:** 3
**Presentation:** 3
**Contribution:** 3
**Rating:** 6
**Confidence:** 4

**Summary:**

This paper tackles post-hoc calibration for FL using global cwECE, explicitly accounting for user-level DP noise. It explores two families: histogram binning via FedBBQ (server-side weighting and a “single big histogram → merged binners” trick) and federated logit scaling (temperature/vector/matrix) with an order-preserving variant to avoid accuracy loss. In results, FedBBQ provides the best cwECE without DP given sufficient rounds, whereas temperature scaling is the most stable and accuracy-preserving under DP.

**Strengths:**

- Post-hoc calibration is widely deployed in practice, so adapting it to non-IID FL with user-level DP is genuinely useful.
- The proposed FedBBQ and federated scaling methods are cleanly designed, easy to implement, and empirically effective.
- The experiments span diverse datasets and baselines and surface actionable insights for research and deployment—for example, binning is sensitive to clipping/noise, while temperature scaling is noise-resilient but can overfit.
- Ablations are thorough, and the code is released (though I didn’t audit it in detail).

**Weaknesses:**

- Lack of rigorous theory. The paper lacks statistical analysis for the federated + DP setting. For FedBBQ, there’s no treatment of bias/variance under user-level clipping + Gaussian noise and the merged-bin ensemble. For FedTemp / FedOPVector / FedOPMatrix, there are no envelope bounds or asymptotics showing when parameter averaging (under data heterogeneity and partial participation) approaches the centralized post-hoc optimum.

- DP tuning is under-specified. Binning methods are clip-sensitive (and thus ε-sensitive) in the DP setting, but there’s no privacy-preserving tuning recipe for choosing $C, C^+$ or rounds $T$.

- Systems reporting is thin. The paper would benefit from concrete bytes/round, client FLOPs, and wall-clock measurements to substantiate the “lightweight” claim and guide deployments.

**Questions:**

- Can you provide statistical analysis for FedBBQ under user-level DP with merged bins (bias/variance, consistency), and a convergence/optimality note for FedTemp / FedOPVector / FedOPMatrix under heterogeneity? To me, averaging a single temperature calibration merely flatten the model's confidence, when does this parameter-averaging strategy work and when doesn't?
- What privacy-preserving recipe do you recommend to tune this hyperparameter, especially for FedBBQ.
- For scaling methods, each client learns its own parameters before aggregation—what do per-client calibration metrics look like pre- and post-federation? Any notable variance across clients?
- Order-preserving guarantees protect top-1 accuracy; do you observe improvements (or regressions) in top-k calibration or ranking metrics (e.g., AUROC, AUPRC) after calibration? Any training strategy or Architecture can preserves top-k ordering?
- Have you evaluated on larger, modern models/datasets (e.g., ViT/ResNet-50)?

---

> ### Author Response · Authors · 2025-11-20
> **Rebuttal 1/2**
>
> We would like to thank the reviewer for their supportive review.
>
> > Systems reporting is thin. The paper would benefit from concrete bytes/round, client FLOPs, and wall-clock measurements to substantiate “lightweight” claim.
>
> Our results on these system metrics were presented in the appendix of the original submission, due to space limitations. In the revised version of the paper we have used the extra page to move some of these results from the appendix into the main paper (see new Table 5) and have highlighted our conclusions in the main text (Section 5).
>
> In more detail, Table 10 & 11 in Appendix A.6 (of the revised paper) contains a computation/communication table that reports per-round communication (bytes exchanged) and client wall-clock time for all proposed methods, alongside the base FL training cost. We emphasise these results show, for all datasets and methods, the additional communication per round due to calibration is orders of magnitude smaller than a single model update in FL training, and the client-side wall-clock time is similarly negligible compared to training the FL model itself.
>
> > What privacy-preserving recipe do you recommend to tune this hyperparameter, especially for FedBBQ?
>
> Our stance is that it suffices to pick these hyperparameters as fixed (data independent) values, since the results are not highly sensitive to this choice.
> To this end, we describe several experiments relevant to hyperparameter tuning in Appendix A.3. To summarise, the current paper shows that all hyperparameters are insensitive as long as they are chosen within a reasonable range and we do not tune them specifically for any of the datasets (i.e., we use fixed values across experiments). In more detail:
> 1. **Number of rounds (T)**: We systematically vary T (Figure 2b) and observe that most methods (especially FedBBQ) perform best in a moderate range (roughly 15–30 rounds), while FedTemp is effective even with T=1. This parameter does not need to be tuned for good performance and setting T=30 gives good utility for the settings and datasets we studied.
> 2. **Clipping norms**: For scaling methods, we found that relatively small norms improve robustness and stability, especially under label skew. These are not sensitive to the choice of $C$ as long as it is not chosen to be extremely large (i.e., > 2).
> For FedBBQ, we study separate clipping norms for positive and negative histograms and identify regimes where cwECE and accuracy are preserved.
> 3. **BBQ bin count (M)**: We vary M (Appendix A.3.3) and show that an intermediate choice (M=7, the value we use in the main experiments) is robust across both non-DP and DP-FL settings. This does not need tuning in the federated setting for good performance.
>
> > For scaling methods, each client learns its own parameters before aggregation—what do per-client calibration metrics look like pre- and post-federation?
>
> To answer your question, in the revised version of the paper we have added a new table in the Appendix (Table 9). For simplicity, we focus on a single run of federated temperature scaling (FedTemp) on CIFAR10 where we vary the heterogeneity parameter $\beta$ (recall smaller beta = more label-skew = more difficult to calibrate).
>
> We present the variance across clients' local ECE using their local temperature parameter $T_k$, the variance across all client temperatures $T_k$ and the average absolute deviation between global optimal temperature $T$ and local $T_k$ i.e., $|T-T_k|$.  We observe there is notable variance across clients, especially when $\beta$ is small. This is expected as these scenarios are more difficult to calibrate because clients have more heterogeneous datasets. This results in large absolute deviations of clients temperature parameters compared to the global optimum. This is true for all calibrators (binning and scaling) and is what motivates us to study solutions that prevent local skew from degrading model accuracy or cwECE.
>
> > Order-preserving guarantees protect top-1 accuracy; do you observe improvements (or regressions) in top-k calibration or ranking metrics (e.g., AUROC, AUPRC) after calibration?
>
> In fact the order-preserving guarantee also protects the full order of predictions.  By construction, the order-preserving method (see Appendix B.6 for the precise details) creates a monotone transformation of the logits that preserves their ordering. Since this preserves the ordering of all predictions, it therefore preserves both the top-1 and top-k accuracies exactly. Likewise for ranking metrics (AUROC/AUPRC), the metric depends only on the ranking of scores, which again is unchanged so these metrics remain unchanged.
>
> We note that in the main paper we only mentioned that order-preserving scaling preserves top-1 accuracy. In our revised paper, we have amended this to also state it preserves top-k accuracy.

---

> > ### Author Response · Authors · 2025-11-20
> > **Rebuttal 2/2**
> >
> > > Have you evaluated on larger, modern models/datasets (e.g., ViT/ResNet-50)?
> >
> > We do not consider these large models.  Our experimental focus is on horizontal FL with resource-constrained devices (e.g., mobile phones), where training and inference for very large models such as ViTs or ResNet-50 end-to-end is still challenging due to overheads. For this regime, we chose architectures that are common and realistic in current FL deployments (e.g., compact CNNs and moderate-size ResNets and LSTMs).
> >
> > Evaluating say ViTs in on-device FL settings is non-trivial and typically requires specialized techniques such as [1], which we view as orthogonal to the federated calibration problem and outside the scope of this work.
> >
> > [1]  *Wu, Meihan, et al. "EFTViT: Efficient Federated Training of Vision Transformers with Masked Images on Resource-Constrained Edge Devices." arXiv preprint arXiv:2412.00334 (2024).*

---

### Author Response · Authors · 2025-11-20
**Revised Paper**

Dear reviewers,

Thank you for your constructive comments. We have uploaded a new revised version of the paper with changes highlighted in red. We have made the following modifications using the allotted extra page to address feedback:
* We have run a new set of experiments (Table 4) and added associated discussion to Section 5, comparing our approaches against 4 train-time methods adapted to the FL setting. We find that our post-hoc calibrators outperform train-time methods, and as observed in prior work, the best calibration occurs when combining train-time with post-hoc. This highlights the importance of studying post-hoc calibration in FL.
* We have moved a version of our computation and communication overhead results to the main paper (Table 5) to highlight the client overheads for post-hoc calibration and added associated discussion to Section 5.
* We have added two new tables to the Appendix (Tables 8 & 9) which display statistics about clients local data as heterogeneity is varied to answer some reviewer comments.

---

### Comment · Area_Chair_cVtp · 2025-11-27

Dear Reviewers,

I noticed that the authors have submitted their rebuttals to your reviews. However, there has been no further engagement or response from your side.

The author-reviewer discussion phase is a critical part of the review process. The authors have put significant effort into addressing your concerns. It is essential that you read their response and acknowledge it.

Please take a moment to:

1. Read the authors' rebuttal carefully.
2. Post a reply indicating whether their response has resolved your concerns.
3. Update your score if appropriate, or explain why your original assessment stands.

Thank you for your immediate attention to this matter.

Best regards,

Area Chair

---

### Author Response · Authors · 2025-12-03
**Summary to AC**

Dear AC,

In summary, the reviewers main concerns with the original submission were primarily (1) presentational/clarity issues which were often answered by information already present in the Appendix, and (2) a lack of comparisons to train-time calibration baselines. We have been able to fully address both concerns using the extra page allowed in our revised paper, with further context and commentary in our rebuttal comments:
* The paper’s presentation has been improved by reorganizing content and providing additional details using the extra space. We have moved a set of experiments on the computation and communication overhead from the Appendix to Section 5, highlighting all post-hoc methods are lightweight and practical for FL, addressing concerns from reviewers (**AaLR, 6xAV**).
* We have added further results to the Appendix which measure various statistics of local clients as we vary heterogeneity, as simulated in our experiments. This highlights that in the most extreme settings there is a very large class-imbalance, and our experiments show our post-hoc methods still obtain positive calibration results in such difficult settings. This addresses concerns from reviewers (**aLWK, JrNY**).
* We have included comprehensive new experiments to cover the missing train-time baseline comparisons. These are a positive result for our work, as they show train-time methods alone are not sufficient for good calibration in FL  In particular, it shows that  1) train-time methods are beaten by our post-hoc FL calibrators and 2) best calibration occurs when combining train-time with post-hoc. These results show the importance of studying post-hoc calibrators in the federated setting, addressing reviewers concerns (**aLWK, 6xAV, JrNY**).

We believe these changes completely address the reviewers’ concerns and have significantly strengthened the paper.

---

### Meta-Review · Area_Chair_8Ukd · 2026-01-06

**Summary:**

**Summary**: This paper addresses the critical problem of post-hoc calibration in Federated Learning (FL), specifically focusing on non-IID data distributions and user-level Differential Privacy (DP). The authors propose two main frameworks: FedBBQ (a federated version of histogram binning) and FedScaling (federated logit scaling, including an order-preserving variant). They evaluate these methods across seven datasets, providing insights into their performance in DP and non-DP settings under varying heterogeneity.


**Strengths**:
1. The problem of post-hoc calibration in non-IID FL with user-level DP is highly relevant and practically important.
2. The proposed FedBBQ and federated scaling methods are well-designed, easy to implement, and empirically effective.
3. The experiments are comprehensive, covering diverse datasets and baselines, and provide actionable insights into the behavior of different calibration methods under DP and heterogeneity.
4.  Code is released


**Weaknesses**:
1. Lack of Theoretical/Formal Analysis (Reviewers AaLR, JRNY, 6xAV): A significant recurring concern is the absence of formal justification for why these mechanisms should work effectively under heterogeneity, and does not provide statistical analysis for the federated learning and DP setting
2. Limited Novelty (Reviewers JRNY and aLWK): Reviewers found the technical contributions to be somewhat limited, primarily integrating established techniques with relatively minor modifications.
3. Need for More Comprehensive Experiments (Reviewers AaLR , JRNY, aLWK, 6xAV): Reviewers ask for comprehensive experiments and analysis, such as systems reporting, comparisons with train-time calibration methods and more recent approaches, discussion of hyperparameter tuning, among others.
4. Clarity Issues (Reviewers 6xAV, aLWK): The paper lacks some key formal definitions, and several technical details are not well justified.
5. Unclear Motivation for Post-hoc Calibration (Reviewer 6xAV)
6. Insufficient Computation and Communication Analysis (Reviewer 6xAV)

**Decision**: Despite the authors' diligent efforts in the rebuttal, which significantly improved the paper's clarity, experimental scope, and addressed many specific questions, fundamental weaknesses persist. The most critical unresolved issue remains the lack of rigorous theoretical and formal analysis for the proposed mechanisms. This was a consistent concern raised by multiple reviewers, and the authors did not provide any such analysis during the rebuttal phase. Furthermore, while the response attempted to clarify the novelty, the core technical contributions are perceived as primarily integrating and extending established calibration ideas rather than introducing fundamental innovation. This persistent gap in theoretical grounding, coupled with the limited originality of the core contributions, prevents the paper from reaching the threshold for acceptance. Therefore, I recommend rejection.

**Reviewer Concerns:**

**Addressed Concerns:**

1.  **Need for More Comprehensive Experiments** (Reviewers AaLR, JRNY, aLWK, 6xAV): Reviewers requested comprehensive experiments and analysis, including systems reporting, comparisons with train-time calibration methods and more recent approaches, and discussions on hyperparameter tuning. The authors addressed these issues by providing additional experimental results or justifications.

2.  **Clarity Issues** (Reviewers 6xAV, aLWK): The paper lacked some key formal definitions, and several technical details were not well justified. The authors addressed these by improving the paper's presentation, reorganizing content, and providing additional details.

3.  **Unclear Motivation for Post-hoc Calibration** (Reviewer 6xAV): The authors addressed this by providing more justifications.

4.  **Insufficient Computation and Communication Analysis** (Reviewer 6xAV): The authors addressed this by moving a set of experiments on computation and communication overhead from the Appendix to Section 5.

**Unsolved Issues:**

1.  **Lack of Theoretical/Formal Analysis** (Reviewers AaLR, JRNY, 6xAV): A significant recurring concern is the absence of formal justification for why these mechanisms should work effectively under heterogeneity and a lack of statistical analysis for the federated learning and DP setting. No response was provided for these issues.

2.  **Limited Novelty** (Reviewers JRNY and aLWK): Reviewers found the technical contributions to be somewhat limited, primarily integrating established techniques with relatively minor modifications. While authors argued their contributions go beyond simple adaptations, detailing specific non-trivial changes for FedBBQ and the order-preserving variant of FedScaling, these still appear to be "primarily integrating and extending established calibration ideas rather than introducing fundamental innovation."

**Reviewer Scores:**

Reviewer AaLR (6 → likely unchanged or decreased, as no response for the statistical analysis), Reviewer JRNY (4 → unchanged), Reviewer aLWK (4 → unchanged), Reviewer 6xAV (2 → unchanged)

---

### Decision · Program_Chairs · 2026-01-26

Reject